# COAT: Compressing Optimizer states and Activation for Memory-Efficient FP8 Training

**Haocheng Xi[1]\*, Han Cai[2], Ligeng Zhu[2], Yao Lu[2], Kurt Keutzer[1], Jianfei Chen[4]†, Song Han[2,3]†**

[1] University of California, Berkeley    [2] NVIDIA    [3] MIT    [4] Tsinghua University
https://github.com/NVlabs/COAT & https://nvlabs.github.io/COAT/

## Abstract

FP8 training has emerged as a promising method for improving training efficiency. Existing frameworks accelerate training by applying FP8 computation to linear layers while leaving optimizer states and activations in higher precision, which fails to fully optimize memory usage. This paper introduces COAT (**C**ompressing **O**ptimizer States and **A**ctivations for FP8 **T**raining), a novel FP8 training framework designed to significantly reduce memory footprint when training large models. COAT addresses current limitations through two key innovations: (1) **Dynamic Range Expansion**, which aligns optimizer state distributions more closely with the FP8 representation range, thereby reducing quantization error, and (2) **Mixed-Granularity Activation Quantization**, which optimizes activation memory using a combination of per-tensor and per-group quantization strategies. Experiments demonstrate that COAT effectively reduces end-to-end training memory footprint by 1.54× compared to BF16 while achieving nearly lossless performance across various tasks, such as Large Language Model pretraining and fine-tuning and Vision Language Model training. COAT also achieves a 1.43× end-to-end training speedup compared to BF16, performing on par with or surpassing TransformerEngine's speedup. COAT enables efficient full-parameter training of large models on fewer GPUs, and facilitates doubling the batch size in distributed training settings, providing a practical solution for scaling large-scale model training. The code is available at https://github.com/NVlabs/COAT.

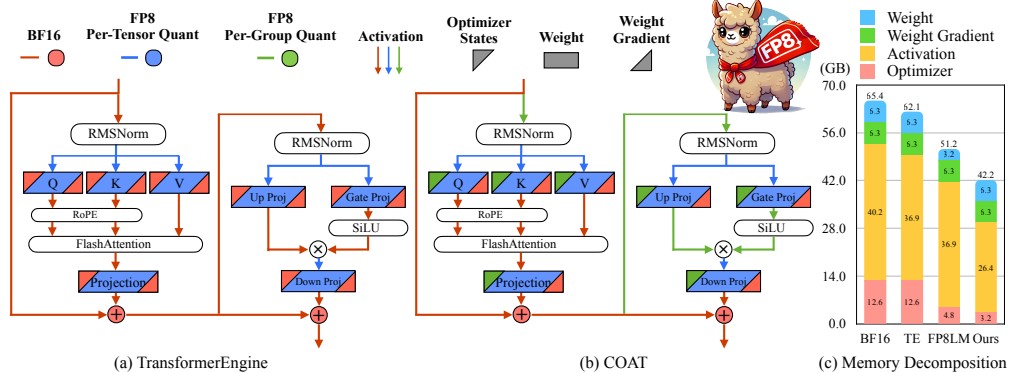

Figure 1: (a,b) Comparing the quantization flow of Transformer Engine and COAT. Both the optimizer states and activations are quantized to FP8 in COAT. (c) End-to-end per-GPU memory comparison when training Llama-2-13B on 8×80G H100 using FSDP.

---

*Part of the work done during an internship at NVIDIA.
†Corresponding author

# 1 INTRODUCTION

Foundation Models (FMs), such as Large Language Models (LLM) and Vision Language Models (VLM), have made significant breakthroughs in various tasks such as reasoning, understanding, and summarization (Dubey et al., 2024; Adler et al., 2024; Team et al., 2024; Lin et al., 2024). However, the training of such models, which often comprise billions of parameters, demands substantial computational resources and memory. This presents substantial challenges, making the training of these foundation models very challenging (Smith et al., 2022; Hoffmann et al., 2022).

Low-precision training has emerged as a promising approach to make FMs training more efficient (Micikevicius et al., 2017; Wang et al., 2018; Zhu et al., 2020; Xi et al., 2023; Wortsman et al., 2023; Xi et al., 2024). By quantizing tensors used in deep neural networks into lower precision, low-precision training effectively speed up the training process and reduce the memory footprint. Currently, BF16 training (Kalamkar et al., 2019; Micikevicius et al., 2017) is the most prevalent low-precision method, and is widely adopted in large-scale training frameworks like DeepSpeed (Rasley et al., 2020) and Megatron-LM (Shoeybi et al., 2019).

With the advent of Nvidia's H100 GPU (NVIDIA, 2024a), FP8 training Micikevicius et al. (2022) is emerging as the next-generation low-precision technique. Compared to BF16, FP8 training has the potential to (1) double the speed and (2) halve the memory footprint. To achieve practical speedup, Transformer Engine (NVIDIA, 2024b) performs matrix multiplications in FP8 precision, leading to faster training. Transformer Engine's memory footprint can be further improved by reducing optimizer states, gradients, weights, and activations to lower precision. As illustrated in Figure 1, FP8-LM (Peng et al., 2023) advances this by further quantizing the gradients, weight master copy, and first-order momentum into FP8. This reduces memory and communication overhead, partially improving memory efficiency. However, they do not tackle the memory consumption of *activations* and still leave the optimizer's second-order momentum in higher precision. The memory problem of activations becomes even more critical when optimizer, gradient, and weights are sharded across multiple GPUs using ZeRO or FSDP. Besides, second-order momentum is more sensitive to quantization than first-order momentum (Fishman et al., 2024), and activations' large spikes also make them hard to quantize to FP8 (Yang et al., 2024). This potential accuracy degradation makes them missing a crucial opportunity to optimize memory further.

In this work, we propose COAT: **C**ompressing **O**ptimizer states and **A**ctivations for memory-efficient FP8 **T**raining to address the aforementioned issue. *COAT significantly reduces the overall memory footprint by quantizing optimizer states and activations into FP8.* For optimizer states, we observe that FP8 format's *representation range is under-utilized* when quantizing them, as illustrated in Figure 2(a). To address this, we introduce a novel *Dynamic Range Expansion* method which adjusts the distribution of optimizer states to better fit within the FP8 range, thereby minimizing quantization error. For activations, we propose *Mixed-Granularity Activation Quantization* to achieve efficient and accurate quantization. We apply fine-grained quantization to non-linear layers and apply per-tensor quantization to linear layers. Per-tensor quantization for matrix multiplications is more efficient and better suited for TensorCores, while fine-grained quantization helps maintain accuracy. These two approaches tackle high memory consumption while ensuring minimal performance degradation. We provide an overview of COAT in Figure 1(b) for demonstration.

We demonstrate the accurate performance of COAT on a wide range of tasks, including LLM pre-training, LLM fine-tuning, and VLM training. COAT achieves nearly lossless performance on all of these tasks. For efficiency results, COAT achieves $1.54\times$ end-to-end memory reduction compared with BF16, and $1.43\times$ end-to-end training speed up on Llama 7B, 13B, and 30B models compared to BF16. COAT also *doubles the batch size* in all realistic distributed training settings, which is crucial for higher speedup and support for longer context length, leading to a more efficient training process for large-scale models.

# 2 RELATED WORK

**Low-precision Training** Low precision training (Wang et al., 2018; Chen et al., 2020; Lin et al., 2022; Wortsman et al., 2023; Xi et al., 2024) has become a prominent technique in modern deep learning, offering reductions in both computational costs and memory requirements. FP16 (half-precision) training (Micikevicius et al., 2017) is the most prevalent low-precision method nowadays.

It introduces loss scaling to address FP16's narrower representation range problem. BF16 training (Kalamkar et al., 2019) refines this approach, as BF16 has a larger representation range and is more stable for large-scale training. In these approaches, forward and backward passes are computed in FP16 or BF16 precision, while master weights, gradients, and optimizers are stored in FP32.

FP8 training (Fishman et al., 2024; Micikevicius et al., 2022) aims to push these efficiency gains further. With the introduction of Nvidia's Hopper GPU architecture, FP8 is emerging as a practical datatype for next-generation low-precision training. Nvidia's Transformer Engine (TE) (NVIDIA, 2024b) is the first framework designed for FP8 mixed-precision training that employs FP8 Tensor-core for linear layer calculation. FP8-LM (Peng et al., 2023) extends FP8 quantization to gradients and optimizer states, further improving the training throughput. However, they fail to reduce the memory usage of activations stored for the backward pass using FP8, and leave second-order momentum in FP16, limiting the full potential of FP8's memory advantages.

**Memory Efficient Optimizers**   While 32-bit optimizer (Kingma, 2014; Loshchilov, 2017) states are widely adopted, several research efforts have been made to reduce the memory footprint of optimizer states through quantization. The 8-bit Adam (Dettmers et al., 2021) introduces a novel data format called dynamic exponent (DE) for quantization, but the adoption of this new data format limits its flexibility. 4-bit Optimizer (Li et al., 2024) further pushes the limit of optimizer quantization to 4-bit by addressing the zero-point problem, but is restricted to fine-tuning tasks. FP8-LM (Peng et al., 2023) quantizes the first-order momentum to FP8 while leaving second-order momentum in FP16, which limits the overall memory savings. (Fishman et al., 2024) finds that second-order momentum is more sensitive to quantization, and proposes to quantize it using E5M2 format.

In addition to quantization, there are other approaches that aim to reduce the memory footprint of the optimizer states (Shazeer & Stern, 2018; Anil et al., 2019; Chen et al., 2024; Zhao et al., 2024), such as low-rank decomposition and optimizer simplification that only store the first-order momentum. These methods are orthogonal to our approach.

## 3   PRELIMINARIES

**FP8 Quantization**   Quantization compresses a high-precision tensor to low-precision to achieve speedup and save memory footprint, at the cost of lower precision and lower representation range. *FP8 format* consists of two encodings - E4M3 and E5M2 (Open Compute Project, 2023). E4M3 has higher precision, while E5M2 has a larger representation range. We define E4M3's min and max values as $\Delta_{\min}^{\text{E4M3}} = 2^{-9}$ and $\Delta_{\max}^{\text{E4M3}} = 448$, while E5M2's min and max values are $\Delta_{\min}^{\text{E5M2}} = 2^{-16}$ and $\Delta_{\min}^{\text{E5M2}} = 57344$. To *quantize* an FP32 tensor $X$ into E4M3 precision, we use a quantizer $Q(\cdot)$ to map the tensor into FP8's representation range. This process can be formulated as

$$X_{\text{FP8}}, S_X = Q(X_{\text{FP32}}), \text{ where } X_{\text{FP8}} = \left\lceil \frac{X_{\text{FP32}}}{S_X} \right\rfloor, \ S_X = \frac{\max\left(|X_{\text{FP32}}|\right)}{\Delta_{\max}^{\text{E4M3}}},$$

where $S_X$ is the scaling factor, $\lceil \cdot \rfloor$ refers to round-to-nearest. Quantize into E5M2 follows a similar procedure, where we only replace $\Delta_{\max}^{\text{E4M3}}$ with $\Delta_{\max}^{\text{E5M2}}$. To map the quantized tensor back to FP32 precision, the *dequantize* operation $DQ(\cdot)$ can be formulated as $X_{\text{FP32}} = DQ(X_{\text{FP8}}, S_X) = S_X X_{\text{FP8}}$.

**Optimizer Update Rule**   Optimizers are widely used in deep learning to update parameters. The most common gradient-based optimizer is Adam/AdamW (Kingma, 2014; Loshchilov, 2017), which uses first-order $m$ and second-order momentum $v$ to achieve better convergence. The update rule of AdamW at time step $t$ can be formulated as:

$$m_t = \beta_1 m_{t-1} + (1 - \beta_1) g_{t-1} \qquad v_t = \beta_2 v_{t-1} + (1 - \beta_2) g_{t-1}^2$$
$$\hat{m}_t = \frac{m_t}{1 - \beta_1^t} \qquad\qquad\qquad \hat{v}_t = \frac{v_t}{1 - \beta_2^t}$$
$$w_{t+1} = w_t - \eta \left( \frac{\hat{m}_t}{\sqrt{\hat{v}_t} + \epsilon} + \lambda w_t \right) \tag{1}$$

where $m_t$ is the first-order momentum, $v_t$ is the second-order momentum, $g_t$ is the gradient, $\beta_1$ and $\beta_2$ are betas of AdamW, $\eta$ is learning rate, $\lambda$ is weight decay, and $\epsilon$ is used to prevent NaN or Inf.

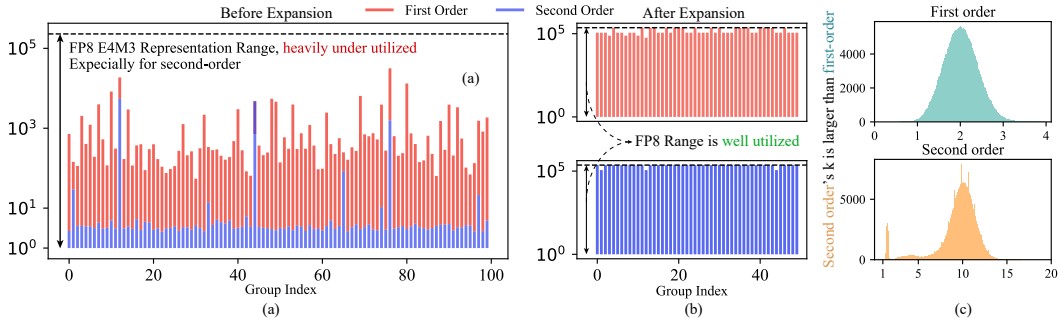

Figure 2: (a) Visualization of optimizer states' dynamic range under per-group quantization. FP8 E4M3's representation range is under-utilized in this case. (b) After dynamic range expansion, FP8's representation range is well utilized. (c) Distribution of $k$ for optimizer states. The second order's $k$ is larger than the first order's $k$, since the second-order momentum's dynamic range is smaller.

To perform **optimizer state quantization**, we adopt per-group quantization for both first-order and second-order momentum, following previous works (Dettmers et al., 2021; Li et al., 2024). Every consecutive $G$ element forms a group ($G$ is defined as the group size), and each group is quantized independently with its own statistics. Optimizer states are stored in FP8 precision, while its scaling factor is stored in BF16. In FSDP or ZeRO-3, the statistics related to optimizer states (e.g. scaling factor) are not synchronized across GPUs, since each GPU maintains its own shard of optimizer states. More details can be found in Appendix A.

## 4 DYNAMIC RANGE EXPANSION FOR ACCURATE OPTIMIZER QUANTIZATION

In subsequent sections, we explain how COAT utilizes FP8 quantization to achieve memory-efficient FP8 training without compromising accuracy. Section 4 focuses on optimizer states quantization, while Section 5 discusses activation quantization.

### 4.1 UNDERSTANDING THE ISSUE OF CURRENT OPTIMIZER STATES QUANTIZATION METHOD

Under per-group quantization, we find that one significant drawback of current quantization methods is that, they can not *fully utilize the representation range* of FP8 and therefore lead to a large quantization error. Take the E4M3 data format as an example, the ratio between E4M3's maximum representable value and minimum representable value is $\Delta_{max}^{E4M3}/\Delta_{min}^{E4M3} = 448 \div \frac{1}{512} = 229376 \approx 2 \times 10^5$. Therefore, for a quantization group $X$, if we want to fully utilize the 256 representable value[1] of FP8, we hope the dynamic range of the quantization group $X$ should cover the entire span between $\Delta_{min}^{E4M3}$ and $\Delta_{max}^{E4M3}$.

To make it more formally, we define *dynamic range* as the ratio between the maximum absolute value and the minimum absolute value within a quantization group $X$:

**Definition 1 (dynamic range)** *Given a set that consists of $G$ real numbers $X = \{x_1, x_2, \ldots, x_G\}$, the dynamic range $\mathcal{R}$ is defined as*

$$\mathcal{R}_X = \frac{\max(|x_1|, |x_2|, \ldots, |x_G|)}{\min(|x_1|, |x_2|, \ldots, |x_G|)},$$

*where $|\cdot|$ denotes the absolute value.*

That is to say, E4M3's dynamic range is $\mathcal{R}_{E4M3} = 448 \times 512 = 229376 \approx 2 \times 10^5$. However, in practice, many quantization groups within the optimizer states fail to effectively map values across this wide range. We observe that optimizer states are highly sparse, with fewer than 1% of values having large magnitudes, while the majority are relatively small and closely clustered. Most groups exhibit low dynamic ranges since large values are so few. As visualized in Figure 2(a), the dynamic range for first-order momentum is typically less than 1e4, and for second-order momentum, it is

---

[1]Actually among them a small amount of values represents NaN and Inf.

Table 1: Quantization error of $\frac{m}{\sqrt{v}}$ under different quantization settings. +Expand means applying our Dynamic Range Expansion method.

| MSE of $\frac{m}{\sqrt{v}}$ | Second Order | | | |
|---|---|---|---|---|
| First Order | E4M3 | E4M3+Expand | E5M2 | E5M2+Expand |
| E4M3 | **20.10** | 18.08 | **25.65** | 18.16 |
| E4M3+Expand | 15.13 | **12.31** | 21.96 | **12.43** |
| E5M2 | **37.02** | 35.96 | **40.30** | 36.00 |
| E5M2+Expand | 17.79 | 15.48 | 23.84 | 15.57 |

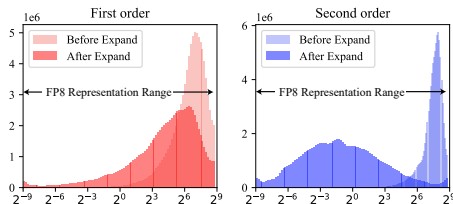

Figure 3: Dynamic Range Expansion can better utilize E4M3 representation range.

usually less than 1e1—both far below the available range of FP8. As a result, a substantial portion of FP8's representational capacity is wasted, leading to a large quantization error.

## 4.2 DYNAMIC RANGE EXPANSION

To address this problem, we introduce a *expand function* $f(\cdot)$ before quantization to expand the dynamic range of the quantization group and align it with E4M3, which can be formalized as $X_{\text{FP8}}, S_X = Q(f(X_{\text{FP32}}))$. The expand function we use is defined as

$$f(x) = \text{sign}(x)|x|^k,$$

where $k$ is used to control the strength of the expansion. In Appendix. F we prove that it is optimal.

For a quantization group $X$, after applying the expand function $f$ to $X$, the dynamic range becomes

$$\mathcal{R}_{f(X)} = \frac{\max(|f(X)|)}{\min(|f(X)|)} = \frac{\max(|\text{sign}(X)X^k|)}{\min(|\text{sign}(X)X^k|)} = \left(\frac{\max(|X|)}{\min(|X|)}\right)^k = (\mathcal{R}_X)^k.$$

Therefore, when $k > 1$, $\mathcal{R}_X$ will be enlarged and become closer to the ideal $\mathcal{R}_{\text{E4M3}}$. The optimal $k$ satisfy that $(\mathcal{R}_X)^k = \mathcal{R}_{\text{E4M3}}$, which means that $k = \log_{\mathcal{R}_X}(\mathcal{R}_{\text{E4M3}})$. With this optimal $k$, $f(X)$ can fully utilize the representation range of E4M3, while the original $X$ can only utilize a small portion of it. As shown in Figure 2(c), the second-order momentum typically has a larger $k$ value ($5 \sim 15$) compared to the first-order momentum's $k$ ($1 \sim 3$). This corresponds to our previous observation, that second-order momentum usually has a smaller dynamic range compared with first-order momentum, so it requires a larger $k$ to align well with E4M3's dynamic range.

We calculate $k$ on-the-fly for every optimizer step and for every quantization group for accuracy consideration. When dequantize, we apply the inverse of expand function $f^{-1}(x) = x^{\frac{1}{k}}$ after dequantization to recover its original value, which can be expressed as $X^{\text{FP32}} = f^{-1}(DQ(X_{\text{FP8}}, S_X))$.

We apply regular quantizer and our dynamic range expansion method to both the first-order momentum $m$ and second-order momentum $v$. As visualized in Figure 3(b), the distribution after expansion can fully utilize the FP8 (E4M3) representation range, which proves the effectiveness of our method. We further quantify the effectiveness of our method in Table 1. In AdamW optimizer step, as stated in Eq. 1, $\frac{m}{\sqrt{v}+\epsilon}$ is the actual effective term for weight update, so we report the MSE of $\frac{m}{\sqrt{v}+\epsilon}$ to quantify the performance of a quantization method. We find that E4M3 is more suitable for first-order momentum than E5M2. For second order momentum, although E4M3 better than E5M2, their quantization error is nearly the same after applying our expand function. Our Dynamic Range Expansion can effectively reduce the MSE by $1.63\times$. Appendix A provide more results.

## 5 MIXED-GRANULARITY ACTIVATION QUANTIZATION

### 5.1 DECOMPOSE THE ACTIVATION MEMORY FOOTPRINT

In the forward pass of neural networks, activations must be preserved for the backward pass to calculate gradients. As illustrated in Table 2, non-linear layers such as LayerNorm/RMSNorm (Ba, 2016; Zhang & Sennrich, 2019) and Activation Functions (Hendrycks & Gimpel, 2016; Shazeer, 2020)

Table 2: Activation memory footprint of different operators. U is a unit to measure memory usage, where 1U = Batch Size × Sequence Length × Hidden Size × 2 bytes (for BF16). For Llama-style model, Act Func refers to SiLU & Multiply, and Linear refers to the summation of QKV/Attn/Up/Gate/Down projection. RMSNorm is upcast to float32 in `transformers` implementation, so the memory usage of LayerNorm in BF16 is 4U. Our method reduces activation memory by quantizing them to FP8. More details about FlashAttention in Appendix D.

| | | Non-Linear | | Attention | | | | Reduction Ratio | |
| | | RMSNorm | Act Func | RoPE | FlashAttn | Linear | Total | Ideal | Achieved |
|---|---|---|---|---|---|---|---|---|---|
| | BF16 | 4U | 8U | 2U | 3U | 5.66U | 22.66U | 1.00× | 1.00× |
| Llama-style | TE | 2U | 8U | 2U | 3U | 3.33U | 18.33U | 1.23× | 1.20× |
| | COAT | 1U | 4U | 2U | 3U | 3.33U | 13.33U | 1.69× | 1.65× |

typically account for approximately 50% of the memory footprint in the Llama model series Touvron et al. (2023). In contrast, linear layers contribute less than 25%. Therefore, it is essential to optimize both linear and non-linear layers to reduce activation memory footprint.[2]

One straightforward approach to achieve this is to quantize the activations to FP8 format prior to each non-linear and linear layer and only save the quantized tensors for backward. However, this introduces significant overhead due to the additional quantization step. Furthermore, non-linear layers can be sensitive to quantization. For example, GLU activation will amplify the spike in activations, resulting in significant quantization errors if not carefully handled (Yang et al., 2024; Fishman et al., 2024). This necessitates us to develop efficient and accurate quantization techniques.

## 5.2 Mixed granularity FP8 precision flow

To address the inefficiency and inaccurate problem, we propose to use **mixed granularity FP8 precision flow** to improve the accuracy without introducing too much overhead. FP8 precision flow requires the input and output of all linear and non-linear layers in FP8. By directly saving the input tensor in FP8 format for the backward pass, we eliminate the need for an extra quantization operation, which reduces the associated overhead. However, this method still suffers from accuracy degradation and necessitates further refinement.

We propose to vary the quantization granularity across different layers to balance precision and efficiency in a *mixed-granularity* manner. For non-linear layers, VS-Quant (Dai et al., 2021) or Per-Block Quant (Xi et al., 2024) methods are well-suited due to their fine-grained and precise nature. For linear layers, we apply per-tensor quantization to maximize the performance of Tensor Cores.[3]

We observe that quantizing the input of layernorm across multiple token axes is detrimental to accuracy. As illustrated in Figure 4(a), when the number of elements that share a scaling factor is fixed, the quantization error increases significantly when quantization is performed across the token axis. Therefore instead of using per-block quantization with block size $B \times B$ as proposed in (Xi et al., 2024), we propose to use per-group quantization with group size $1 \times G$, where $G = B^2$ to keep the granularity the same. This approach enhances the accuracy of non-linear layers while maintaining efficiency. Our precise FP8 precision flow is visualized in Figure 1(a), where we display the full precision flow for a Llama-style decoder layer, both forward and backward pass.

We also propose Group Scaling, an efficient per-tensor scaling method that balances the performance and precision. To perform per-tensor quantization, the maximum absolute value of the tensor needs to be calculated through max reduction, adding a lot of overhead.

In our Group Scaling, we address these problems by splitting the max reduction into two stages: (1) performing max reduction on each $1 \times G$ element and storing the results as intermediate values; (2) applying max reduction on the intermediate tensor to obtain the per-tensor max value. The first stage can be seamlessly fused with the previous operation, adding minimal overhead, while the second stage is more efficient than doing max reduction on the entire tensor, as the intermediate result is $G\times$ smaller than the original tensor. As illustrated in Figure 4(b), Group Scaling successfully reduces the max reduction overhead compared with just-in-time scaling.

---

[2]RoPEs and FlashAttentions are more as a distinct module and should be handled separately.

[3]Finer-grained methods can also be applied since they are generally compatible with FP8 precision flow.

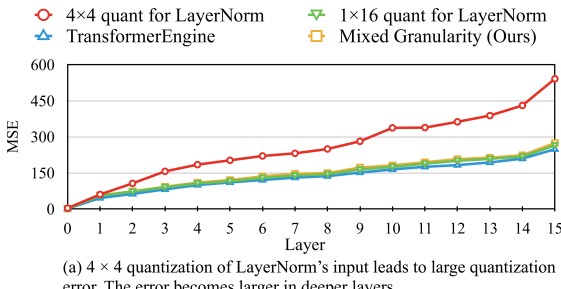

| Tensor Size | Just-in-Time Scaling | | Group Scaling (Ours) |
|---|---|---|---|
| 11008×16384 | 1.37 ms | 14× | 0.10 ms |
| 11008×8192 | 0.89 ms | 11× | 0.08 ms |
| 4096×16384 | 0.55 ms | 8× | 0.07 ms |
| 4096×8192 | 0.32 ms | 5× | 0.06 ms |

(a) 4 × 4 quantization of LayerNorm's input leads to large quantization error. The error becomes larger in deeper layers.

(b) Time usage to calculate per-tensor max value. Our Group Scaling greatly reduce the reduction overhead, while it doesn't suffer from Delayed Scaling's instability issue.

Figure 4: (a) Quantization Error in forward pass. (b) Time comparison of various scaling methods.

In comparison, TransformerEngine proposes delayed scaling to avoid the on-the-fly max reduction required for per-tensor quantization, and can also fuse the division process into the previous operator to optimizes memory accesses. However, we find that using the current tensor's statistics to compute the scaling factor is no worse, and could be even better in precision, than using the delayed scaling heuristic. Therefore, we advocate for Group Scaling as a simpler and more flexible alternative that can potentially offer improved numerical stability and precision while not being significantly slower than Delayed Scaling.

# 6 EXPERIMENTS

## 6.1 ACCURACY EXPERIMENTS

**Setups** We compare COAT with BF16 training and TransformerEngine (NVIDIA, 2024b) baselines. For all experiments, we adopt the default hyperparameters in the official training recipe. We validate the effectiveness of our method on multiple tasks, including Large Language Model (LLM) pretraining and fine-tuning, and Vision Language model (VLM) training. For LLM pertaining, we report the perplexity on Wikitext 103 (Merity et al., 2016), C4 (Raffel et al., 2020), and Pile (Gao et al., 2020), and the accuracy on COPA (Gordon et al., 2012), ARC (Clark et al., 2018), SciQ (Welbl et al., 2017), and HellaSwag (Zellers et al., 2019). For LLM fine-tuning, we conduct experiments in math corpus, and evaluate on Mathmeticas (Davies et al., 2021), SVAMP (Patel et al., 2021), NumGLUE (Mishra et al., 2022), and GSM8K (Cobbe et al., 2021). For VLM training, we report the score on VideoMME (Fu et al., 2024), POPE (Li et al., 2023b), VizWiz (Gurari et al., 2018), GQA (Hudson & Manning, 2019), VQAv2 (Goyal et al., 2017), TextVQA (Singh et al., 2019), SEED (Li et al., 2023a), and MMMU Validation Set (Yue et al., 2024). We use $1 \times 128$ per-group quantization for optimizer states and $1 \times 16$ per-group quantization for non-linear layer activations.

### 6.1.1 LLM PRETRAINING

To evaluate our method when pretraining LLMs, We train OLMo-1B (Groeneveld et al., 2024) and OLMo-7B on Dolma (Soldaini et al., 2024). Following the official report, we use a global batch size of 4M tokens (2048 macro batch size, with a sequence length of 2048 tokens). We use PyTorch FSDP in our experiments.

For OLMo-1B, we conduct pretraining for 300B tokens, which corresponds to 75k training steps. We report the training curve in Figure 5, and report the perplexity and accuracy result in Table 3. The training curve and downstream task performance were consistent with BF16 training baseline, validating the effectiveness of COAT. For OLMo-7B experiment, we pretrain for 250B tokens, which corresponds to 40k training steps.we report the training curve in Figure 6 and downstream task performance in Table 4, where COAT also aligns well with the baseline and is nearly lossless.

### 6.1.2 LLM FINE-TUNING

We further evaluate our method on LLM fine-tuning task. We focus on math corpus and fine-tune a Llama-2-7B model on the MAmmoTH (Yue et al., 2023) dataset. We train for 3 epochs, and report the downstream tasks performance in 5. After fine-tuning, COAT still performs consistently with the baselines on the downstream task performance, proving the accurateness of our method.

Table 3: OLMo-1B pretraining performance on downstream tasks. We report the performance after training for 300B tokens.

|        | Train Loss | WikiText | C4     | Pile   | Avg ppl |
|--------|------------|----------|--------|--------|---------|
| BF16   | 2.551      | 15.234   | 15.538 | 10.563 | 10.083  |
| COAT   | 2.568      | 15.384   | 15.695 | 10.672 | 10.176  |
|        | COPA       | ARC(Easy)| SciQ   | HellaSwag | Avg Acc |
| BF16   | 77.0 %     | 57.3 %   | 84.0 % | 54.5 %    | 68.2 %  |
| COAT   | 75.0 %     | 58.1 %   | 83.9 % | 54.3 %    | 67.8 %  |

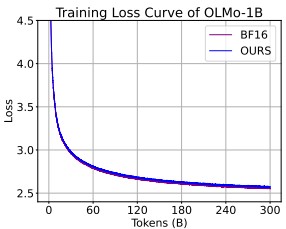

Figure 5: OLMo-1B training loss curve.

Table 4: OLMo-7B pretraining performance on downstream tasks. We report the performance after training for 250B tokens.

|        | Train Loss | WikiText | C4     | Pile   | Avg ppl |
|--------|------------|----------|--------|--------|---------|
| BF16   | 2.366      | 12.053   | 12.874 | 8.596  | 11.174  |
| COAT   | 2.379      | 12.166   | 12.988 | 8.684  | 11.279  |
|        | COPA       | ARC(Easy)| SciQ   | HellaSwag | Avg Acc |
| BF16   | 83.0%      | 65.7%    | 87.5%  | 56.9%     | 73.2 %  |
| COAT   | 81.0%      | 61.9%    | 87.2%  | 60.6%     | 72.7 %  |

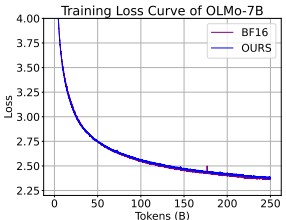

Figure 6: OLMo-7B training loss curve.

### 6.1.3 VLM TRAINING

We also evaluate COAT on vision language models. We conduct experiments on VILA (Lin et al., 2024) and perform stage-3 SFT of VILA1.5-7B using the same SFT data mixture employed by VILA's original paper (Chen et al., 2023; Xu et al., 2024) and set the global batch size to 1024. We pad the sequence length to a multiple of 4 for efficiency consideration. The training loss curve is visualized in Figure 7. We report the downstream task performance and their average in Table 6. We find that COAT performs on par with BF16 training and is better than the TransformerEngine baseline, which demonstrates the accurateness of our method. We further visualize the VLM captioning experiment in Appendix B to prove the effectiveness of COAT on generation tasks.

### 6.2 MEMORY SAVING AND SPEEDUP

We test the memory saving and speedup result of COAT in two settings: The results on a single transformer layer help to accurately analyze the capabilities for memory saving and speedup, while the end-to-end results reflect the practical benefits of our method in real-world applications.

### 6.2.1 MEMORY SAVING AND SPEEDUP FOR A SINGLE TRANSFORMER LAYER

Table 7 highlights the speedup and memory reduction achieved for a single transformer layer. We conducted experiments with a batch size of 4, varying the hidden sizes between 2048 and 4096, and sequence lengths of 2048 and 4096.

COAT demonstrates better speedup compared with TE, and significantly better memory reduction ability compared with BF16 and TE. Our approach achieves up to $1.57\times$ speedup over BF16 and achieves a consistent $1.65\times$ memory reduction compared to BF16, which is very close to the theoretically $1.69\times$ reported in Table 2. The speedup ratio becomes larger with larger hidden sizes and longer sequence lengths.

### 6.2.2 SPEEDUP AND MEMORY SAVING FOR END-TO-END TRAINING

Table 8 presents a detailed comparison of end-to-end memory reduction and speedup results across different configurations for transformer models, specifically Llama-2-7B, Llama-2-13B, and Llama-30B, with variations in the number of GPUs used (1, 2, 4, and 8). It highlights COAT's effectiveness

Table 5: Evaluation result of fine-tuning Llama-2-7B on math corpus. `Llama-2-7B` refers to the evaluation metric before fine-tuning. TE refers to TransformerEngine.

|  | **Mathmeticas** | **SVAMP** | **NumGLUE** | **GSM8k** | **Avg** |
|---|---|---|---|---|---|
| `Llama-2-7B` | 6.0 | 14.6 | 34.5 | 29.9 | 21.3 |
| BF16 | 46.3 | 64.2 | 54.8 | 57.7 | 55.7 |
| TE | 45.3 | 66.1 | 53.5 | 57.7 | 55.6 |
| COAT | 47.8 | 64.4 | 53.3 | 56.6 | 55.5 |

Table 6: VILA1.5-7B Stage-3 SFT performance on downstream tasks. * means it has seen the training data.

| Stage 3 | VideoMME | POPE | VizWiz | GQA* | VQAv2* |
|---|---|---|---|---|---|
| BF16 | 42.96 | 86.90 | 61.42 | 64.55 | 81.47 |
| TE | 43.19 | 87.64 | 57.61 | 64.53 | 81.34 |
| COAT | 44.56 | 87.43 | 61.36 | 64.44 | 81.20 |

| | | SEED | | | |
|---|---|---|---|---|---|
| Stage 3 | TextVQA | Image | Video | MMMU Val | Average |
| BF16 | 65.60 | 73.40 | 45.65 | 38.56 | 62.80 |
| TE | 64.70 | 73.51 | 43.12 | 35.89 | 61.88 |
| COAT | 64.65 | 73.36 | 43.76 | 37.22 | 62.51 |

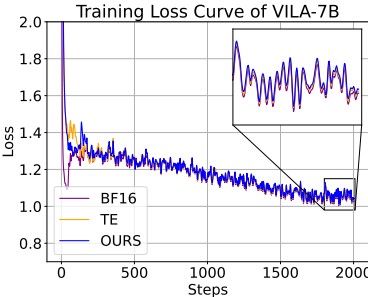

Figure 7: VILA1.5-7B Stage-3 SFT loss curve.

Table 7: Memory Saving and Speedup for a single Transformer Layer. Memory refers to Activation Memory. Our method achieves better speedup than TransformerEngine and significantly reduces the activation memory footprint by $1.65\times$.

| *Hidden Size = 2048, Batch Size = 4* | | | | | | | | | | | | |
|---|---|---|---|---|---|---|---|---|---|---|---|---|
| | | | Sequence Length = 2048 | | | | | | Sequence Length = 4096 | | | |
| | Forward | Backward | Total | Ratio | Memory | Ratio | Forward | Backward | Total | Ratio | Memory | Ratio |
| BF16 | 3.36 | 8.47 | 11.83 | $1.00\times$ | 1457 MB | $1.00\times$ | 6.88 | 17.24 | 24.12 | $1.00\times$ | 2914 MB | $1.00\times$ |
| TE | 2.96 | 5.32 | 8.28 | $1.42\times$ | 1210 MB | $1.20\times$ | 5.94 | 11.29 | 17.23 | $1.39\times$ | 2420 MB | $1.20\times$ |
| COAT | 2.88 | 5.16 | 8.04 | $\mathbf{1.47\times}$ | 883 MB | $\mathbf{1.65\times}$ | 5.89 | 10.82 | 16.71 | $\mathbf{1.44\times}$ | 1766 MB | $\mathbf{1.65\times}$ |
| *Hidden Size = 4096, Batch Size = 4* | | | | | | | | | | | | |
| | | | Sequence Length = 2048 | | | | | | Sequence Length = 4096 | | | |
| | Forward | Backward | Total | Ratio | Memory | Ratio | Forward | Backward | Total | Ratio | Memory | Ratio |
| BF16 | 7.77 | 18.78 | 26.55 | $1.00\times$ | 2914 MB | $1.00\times$ | 16.37 | 38.43 | 54.80 | $1.00\times$ | 5828 MB | $1.00\times$ |
| TE | 6.19 | 11.79 | 17.98 | $1.47\times$ | 2420 MB | $1.20\times$ | 12.66 | 24.58 | 37.24 | $1.47\times$ | 4840 MB | $1.20\times$ |
| COAT | 5.89 | 10.96 | 16.85 | $\mathbf{1.57\times}$ | 1766 MB | $\mathbf{1.65\times}$ | 12.16 | 23.44 | 35.6 | $\mathbf{1.53\times}$ | 3533 MB | $\mathbf{1.65\times}$ |

in reducing end-to-end memory footprint and the speedup compared to standard BF16 and TransformerEngine (TE) setups under varying conditions of batch size and context length.

COAT allows full-parameter training of Llama-2-7B on *a single GPU*, where BF16 and TE both out of memory (OOM). Similarly, for the Llama-2-13B and Llama-30B models, our method enables 2-GPU training for Llama-2-13B and 8-GPU training for Llama-30B when Batch Size = 1.

In all multi-GPU training setting, COAT can double the micro-batch size and therefore lead to even higher speedup. For example, our method can achieve $2.25\times$ speedup when training Llama-2-13B on 4-GPUs since we can effectively increase the batch size to 2.

Overall, COAT significantly reduces end-to-end memory usage by up to $1.55\times$ and speeds up the end-to-end training by nearly $1.44\times$. This facilitates full-parameter training on fewer GPUs, which is particularly beneficial for larger language models.

Table 8: End-to-end memory reduction and speedup results. BS refers to batch size. CL refers to context length. We report token/s per GPU for speed results. ‡ means CL=1024.

| **Llama-2-7B** | | *Context Length = 2048* | | | | *Maximum Batch Size, Context Length = 2048* | | |
|---|---|---|---|---|---|---|---|---|
| | | Optimizer | Activations | Peak | Ratio | Max BS | Speed | Ratio |
| 1 GPU$_{BS=1}$ | BF16 | - | - | OOM | - | - | OOM | - |
| | TE | - | - | OOM | - | - | OOM | - |
| | COAT | 13.1 GB | 8.1 GB | 79.3 GB | ✓ | 1 | 5906 token/s | ✓ |
| 2 GPU$_{BS=2}$ | BF16 | - | - | OOM | - | 1 | 6130 token/s | 1.00× |
| | TE | - | - | OOM | - | 1 | 6842 token/s | 1.11× |
| | COAT | 6.5GB | 16.9 GB | 52.8 GB | ✓ | 4 | 11351 token/s | 1.85× |
| 4 GPU$_{BS=2}$ | BF16 | 13.1 GB | 25.8 GB | 55.1 GB | 1.00× | 2 | 7730 token/s | 1.00× |
| | TE | 13.1 GB | 21.9 GB | 51.1 GB | 1.08× | 2 | 9577 token/s | 1.24× |
| | COAT | 3.2 GB | 16.9 GB | 35.6 GB | 1.54× | 4 | 11257 token/s | 1.45× |
| 8 GPU$_{BS=2}$ | BF16 | 6.5 GB | 25.8 GB | 41.2 GB | 1.00× | 4 | 8238 token/s | 1.00× |
| | TE | 6.5 GB | 21.9 GB | 37.2 GB | 1.11× | 4 | 11704 token/s | 1.42× |
| | COAT | 1.6 GB | 16.9 GB | 27.0 GB | 1.52× | 8 | 11241 token/s | 1.36× |
| **Llama-2-13B** | | *Context Length = 2048* | | | | *Maximum Batch Size, Context Length = 2048* | | |
| | | Optimizer | Activations | Peak | Ratio | Max BS | Speed | Ratio |
| 2 GPU$_{BS=1}^{‡}$ | BF16 | - | - | OOM | - | - | OOM | - |
| | TE | - | - | OOM | - | - | OOM | - |
| | COAT | 12.6 GB | 10.1 GB | 73.2 GB | ✓ | 1 | 2137 token/s | ✓ |
| 4 GPU$_{BS=1}$ | BF16 | 25.1 GB | 20.1 GB | 76.1 GB | 1.00× | 1 | 2345 token/s | 1.00× |
| | TE | 25.1 GB | 17.2 GB | 73.0 GB | 1.04× | 1 | 2851 token/s | 1.21× |
| | COAT | 6.3 GB | 13.2 GB | 49.1 GB | 1.55× | 2 | 5295 token/s | 2.25× |
| 8 GPU$_{BS=1}$ | BF16 | 12.6 GB | 20.1 GB | 49.4 GB | 1.00× | 2 | 3907 token/s | 1.00× |
| | TE | 12.6 GB | 17.2 GB | 46.5 GB | 1.06× | 2 | 5604 token/s | 1.43× |
| | COAT | 3.1 GB | 13.2 GB | 32.5 GB | 1.52× | 4 | 5650 token/s | 1.44× |
| **Llama-30B** | | *Context Length = 2048* | | | | *Maximum Batch Size, Context Length = 2048* | | |
| | | Optimizer | Activations | Peak | Ratio | Max BS | Speed | Ratio |
| 8 GPU$_{BS=1}$ | BF16 | - | - | OOM | - | - | OOM | - |
| | TE | - | - | OOM | - | - | OOM | - |
| | COAT | 7.8 GB | 24.2 GB | 70.5 GB | ✓ | 1 | 1363 token/s | ✓ |

Table 9: Dynamic Range Expansion is compatible with DE8 (8-bit dynamic quantization).

| | Second Order | | | | | |
|---|---|---|---|---|---|---|
| First Order | E4M3 | E4M3 + Expand | E5M2 | E5M2 + Expand | DE8 | DE8 + Expand |
| E4M3 + Expand | 15.13 | **12.31** | 21.96 | **12.43** | 14.01 | 18.84 |
| DE8 | 12.11 | 8.27 | 20.02 | 8.43 | **10.54** | 16.25 |
| DE8 + Expand | 11.57 | **7.47** | 19.69 | **7.65** | 9.91 | 15.81 |

## 6.3 ABLATION STUDIES

### 6.3.1 DYNAMIC RANGE EXPANSION'S COMPATIBILITY WITH OTHER DATA FORMATS

Dynamic Exponent Quantization (DE) was proposed in (Dettmers et al., 2021) to quantize the optimizer states since its representation range has a range of 7 orders of magnitude and is very suitable for optimizer states quantization. Therefore, they proposed to quantize both first-order and second-order momentum with 8-bit DE.

We report the quantization error of $\frac{m}{\sqrt{v}}$ in Table 9, and find that applying our dynamic range expansion method to 8-bit DE can further reduce the quantization error by $1.41\times$. Specifically, the lowest quantization error is achieved when first-order momentum $m$ is quantized with 8-bit DE + Dynamic Range Expansion, and second-order momentum $v$ is quantized with E4M3/E5M2 + Dynamic Range Expansion. This proves the effectiveness of our method.

## 7 CONCLUSION

In this work, we present COAT, a memory-efficient FP8 training framework for foundation models by quantizing both optimizer states and activations into FP8 format. We observe that the FP8 for-

mat's representation range is not fully utilized when quantizing optimizer states, prompting us to propose Dynamic Range Expansion to align their dynamic range. We then identify the importance of quantizing non-linear layers, and propose mixed-granularity FP8 precision flow to quantize the activations accurately without introducing too much overhead.

Extensive experiments on LLM and VLM training and fine-tuning demonstrate that COAT can achieve nearly lossless performance. In end-to-end training, COAT achieves a $1.54$ memory reduction and $1.43$ speedup compared to BF16, and is comparable or even faster to TransformerEngine's training speed. Our method also enables full-parameter training of billion-scale models on fewer GPUs and is able to double the batch size in realistic settings. These results highlight COAT's capability to enable memory-efficient, large-scale model training without sacrificing accuracy, providing a highly effective solution for memory-constrained environments. Future work could further explore combining our proposed approach with other low-precision gradient compression methods to reduce communication overhead.

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

## A    DETAILS ABOUT OPTIMIZER STATES QUANTIZATION

When performing `optimizer.step()`, We first dequantize the optimizer states into FP32, then update the optimizer states and weights in FP32 precision. In the end, we quantize the updated optimizer states back to FP8 and store it in GPU memory. This can be formulated as:

$$m_{t-1} = DQ(m_{t-1}^q, S_{m_{t-1}}) \qquad \text{(Dequantize to FP32)}$$
$$v_{t-1} = DQ(v_{t-1}^q, S_{v_{t-1}}) \qquad \text{(Dequantize to FP32)}$$
$$m_t = \beta_1 * m_{t-1} + (1 - \beta_1) * g_t$$
$$v_t = \beta_2 * v_{t-1} + (1 - \beta_2) * g_t^2$$
$$\hat{m}_t = \frac{m_t}{1 - \beta_1^t}$$
$$\hat{v}_t = \frac{v_t}{1 - \beta_2^t}$$
$$w_{t+1} = w_t - \eta \left( \frac{\hat{m}_t}{\sqrt{\hat{v}_t} + \epsilon} + \lambda w_t \right)$$
$$m_t^q, S_{m_t} = Q(m_t) \qquad \text{(Quantize to FP8)}$$
$$v_t^q, S_{v_t} = Q(v_t) \qquad \text{(Quantize to FP8)}$$

In the FSDP setting, dynamic range synchronization across GPUs is unnecessary due to the unique distributed training architecture. The approach leverages sharding strategies that eliminate the need for cross-GPU synchronization. Each GPU independently handles its own optimizer state shards. Since dynamic range is calculated based on local device statistics and optimizer states remain locally computed and updated, we do not need to synchronize the dynamic range across GPUs.

## B    QUALITATIVE EXAMPLE - VISION LANGUAGE MODEL CAPTIONING

## C    VISUALIZATION OF EXPAND FUCTION

We further visualize Figure 3 by flattening it. This helps to understand the effectiveness of our method since floating point numbers can be more easily understood in a binary manner.

## D    DETAILED EXPLANATION FOR TABLE 2

We mainly explain why the reduction ratio of linear layers is not exactly 50% here. 5.66U comes from: QKV Projection - 1U, Attention Projection - 1U, Up&Gate Projection - 1U, Down Projection - 2.66U. Among them, the attention projection's input is exactly the output of FlashAttention. However, FlashAttention will save its input (QKV) and its output itself, so FlashAttention and Attention Projection actually share the same tensor when saving it. In Python, this tensor will only be saved for 1 time. So we should not further store an FP8 version of the Attention Projection's input (Zhang et al., 2025a; 2024), since this will even increase the memory usage by 0.5U if we do not change the code of FlashAttention. Although we need to quantize the input to FP8 again in backward pass, we can store the scaling factor to reduce this additional overhead.

Therefore, after FP8 quantization, the memory usage of linear layers comes from: QKV Projection - 0.5U, Attention Projection - 1U, Up&Gate Projection - 0.5U, Down Projection - 1.33U. They sum up to 3.33U.

## E    DETAILED EXPLANATION FOR FIGURE 1(C)

We mainly discuss FP8-LM in this section. FP8-LM reduces the gradient communication precision from FP32 to FP8 and therefore greatly reduces the communication overhead. It also reduces the

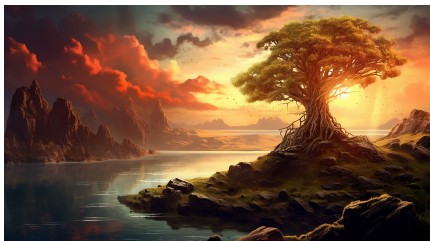

## Instruction: Briefly describe this figure.

**BF16**: The image depicts a fantastical scene set in a mountainous landscape. Dominating the center of the image is a large tree, its twisted and gnarled form standing on a rocky outcropping. The tree is surrounded by a serene lake, its calm waters reflecting the vibrant colors of the sky above. The sky is a spectacle in itself, painted with hues of orange and red as the sun sets behind the mountains, casting a warm glow over the entire scene. The mountains in the background add a sense of depth and grandeur to the image, their peaks reaching up towards the sky. The overall composition of the image suggests a tranquil yet powerful natural setting, possibly inspired by mythology or fantasy.

**COAT**: The image presents a captivating scene of a solitary tree standing on a rocky outcropping. The tree, with its twisted and gnarled form, is a vibrant green, contrasting with the surrounding landscape. The rocky outcropping on which it stands is rugged and uneven, adding to the overall wildness of the scene. The background is a breathtaking view of a mountainous landscape under a sky ablaze with hues of orange and red, possibly indicating a sunrise or sunset. The sun is partially obscured by the clouds, casting a warm glow that illuminates the scene. The image is rendered in a digital art style, which adds a sense of realism and depth to the scene. The artist has skillfully used color and lighting to create a sense of tranquility and solitude, with the tree standing as a solitary figure against the vastness of the landscape. The overall composition of the image is balanced and harmonious, with each element complementing the others to create a cohesive whole.

## Question: Why People love this?

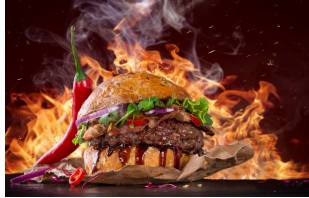

**BF16**: People love this because it is a visually appealing and appetizing image of a delicious burger. The burger is presented on a paper bag, which adds a rustic touch to the presentation. The combination of a juicy burger with a red pepper and a fiery background creates an enticing and mouth-watering scene. The image captures the essence of a classic American favorite, the burger, and the excitement of a freshly cooked meal. The vibrant colors and the dramatic backdrop of the fire make the image even more captivating, which is why it is likely to be appreciated by many people.

**COAT**: People love this image of a burger on a bun with various toppings, such as onions, tomatoes, and lettuce, because it represents a delicious and visually appealing meal. The burger is placed on a piece of paper, which adds an interesting and casual touch to the presentation. The fiery background, with a fire or a flame, creates a dramatic and exciting atmosphere, which can evoke a sense of excitement and anticipation for the meal. The combination of the mouth-watering burger and the striking background can make the image appealing and appetizing to many people, especially those who enjoy burgers and similar dishes.

Figure 8: Comparison of BF16 and COAT on VLM captioning. COAT can accurately summarize the figure and identify the key points in the figure.

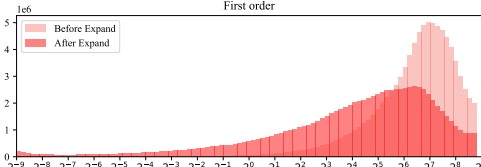 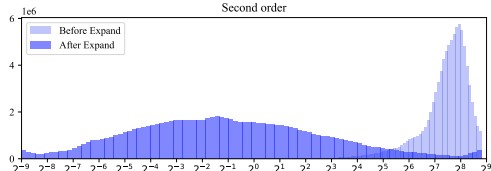

Figure 9: Axis is of base 2.

master weight's precision from FP32 to FP16/BF16, and reduces the optimizer states precision from FP32 to BF16/FP8. For optimizer states and master weights, the memory footprint reduction result is consistent with our bar. However, for gradient, although FP8-LM can quantize the gradient into FP8, it still needs to preserve the main gradient in FP32 for gradient accumulation. Therefore it can not reduce the memory footprint of the weight gradient.

## F    PROOF THE EXPAND FUNCTION IS OPTIMAL

We prove that the choice of $f(x) = x^k$ is optimal given certain basic assumptions for the expansion function $f(x)$.

1. $f(x)$ is an odd function, where $f(x) + f(-x) = 0$, since the representation range of FP8 is symmetric. Therefore, in the following discussion, we only consider the case when $x \geq 0$.

2. $f(x)$ is continuously differentiable.

3. $f(x)$ is monotonically increasing. This is because we want to keep the relative order before and after quantization or expansion.

4. $f(0) = 0$ and $f(1) = 1$. $f(0) = 0$ since $f(x)$ is odd. $f(1) = 1$ since we can always consider $g(x) = \frac{f(x)}{f(1)}$ without changing its expansion ability.

5. For any $x > y > 0, r > 0$, $\frac{f(x)}{f(y)} = \frac{f(rx)}{f(ry)}$. This is used to make sure the function is scale-invariant. For example, if you multiply the unquantized value $X$ by 10, the value after expansion and quantization is the same as if you do not multiply the value by 10.

We prove that based on these assumptions, the only function that meets these assumptions is $f(x) = \text{sign}(x)|x|^k$.

**Proof:** We only consider the case when $x > 0$. We denote $f(2) = m > 1$. From assumption 5, we have for any $p > r > s > q$, if $pq = rs$, then $f(p)f(q) = f(r)f(s)$, since

$$\frac{f(p)}{f(r)} = \frac{f\big(p \cdot (s/p)\big)}{f\big(r \cdot (s/p)\big)} = \frac{f(s)}{f(q)}.$$

And for any $p = rs$,

$$f(p) = f(p)f(1) = f(r)f(s).$$

Therefore, for any $x = 2^t, t \in \mathbb{N}$,

$$f(x) = f(2^t) = f(2)^t = m^t.$$

It is also true that for any $x = 2^{-t}, t \in \mathbb{N}$,

$$f(x) = \frac{f(1)}{f(\frac{1}{x})} = \frac{f(1)}{f(2^t)} = \frac{1}{m^t} = m^{-t}, \text{ so that } f(x) = m^{-t}.$$

Therefore, for any $N \in \mathbb{N}$, $x = 2^t, t = \sum_{i=-N}^{N} c_i \cdot 2^i, c_i \in \{0,1\}$,

$$f(x) = f\Big( \prod_{i=-N}^{N} 2^{c_i \cdot 2^i} \Big) = \prod_{i=-N}^{N} f(c_i \cdot 2^i) = \prod_{i=-N}^{N} m^{c_i \cdot 2^i} = m^{\sum_{i=-N}^{N} c_i \cdot 2^i} = m^t.$$

For $\forall x > 0$, it can be written in the form of $x = 2^t, t \in \mathbb{R}$. Consider the binary representation of $t$, denoted by $t = \sum_{i=-\infty}^{\infty} c_i \cdot 2^i$, where $c_i \in \{0,1\}$. For any integer $N > 0$, there exists two series $\{a_i\}_{-N}^{N}$ and $\{b_i\}_{-N}^{N}$ that satisfies $a_{-N} = 0, b_{-N} = 1$, and $a_i = b_i$ for $-N+1 \le i \le N$,

$$t_a \le t < t_b,$$

where $t_a = \sum_{i=-N}^{N} a_i \cdot 2^i$ and $t_b = \sum_{i=-N}^{N} b_i \cdot 2^i$. Since $f(x)$ is continuous, the values of $f(x)$ at $x_a = 2^{t_a}$ and $x_b = 2^{t_b}$ can be used to approximate $f(x) = f(2^t)$, and by taking the limit as $N \to \infty$, the equality holds:

$$f(x) = \lim_{N \to \infty} f(x_a) = \lim_{N \to \infty} f(x_b).$$

Therefore for $\forall t \in \mathbb{R}$, $f(2^t) = m^t$ for some $m > 0$. Therefore $\forall x > 0$, $f(x) = m^{\log_2(x)}$. Using the exponential and logarithmic identity $m^{\log_2(x)} = x^{\log_2(m)}$, this simplifies to:

$$f(x) = x^{\log_2(m)},$$

where $\log_2(m)$ is a constant determined by $f(2) = m$. If we denote $\log_2(m) = k$, then $f(x) = x^k$.

## G  IMPLEMENTATION DETAILS OF COAT

This section highlights the key differences and details of the implementation of the COAT LLaMA model. The primary focus is on the techniques used to handle quantization and optimization efficiently.

### G.1 MODEL ARCHITECTURE

The COAT LLaMA implementation introduces three new modules, which are not present in the Hugging Face implementation, specifically designed to optimize for FP8 precision flow:

1. **BeforeAttention** module (inherit from torch.autograd.Function) calculates the LayerNorm and QKV Projection before RoPE and correctly handles how quantized inputs interact with the residual connection. If Attention is quantized (Zhang et al., 2025a; 2024; Shah et al., 2024; Zhang et al., 2025b; Xi et al., 2025), then the output of this module should also be quantized to accommodate the input of Attention.

2. **AfterAttention** (inherit from torch.autograd.Function) calculate the attention projection layer and ensures quantized outputs are correctly scaled and summed up with residual connection.

3. **MLPResidual** (inherit from torch.autograd.Function) handles quantization within the feed-forward network (FFN) efficiently. It also incorporates residual connections in this module to handle it efficiently.

These modules ensure that the computation flow after quantization is correct. We fuse the group scaling method into the *add* operator in residual connection, therefore we need to manually control how the gradient flows in the network.

### G.2 TRITON-BASED FP8 KERNELS FOR LINEAR LAYERS

The linear layers utilize Triton to implement custom FP8 matrix multiplication kernels. This implementation introduces the following optimizations:

1. Block Size Tuning: The block sizes for dimensions $M$, $N$, and $K$ are autotuned, with options chosen from $\{128, 256\}$. This dynamic tuning optimizes performance for the specific hardware environment.

2. Group Scaling Technique: The output of linear layer should be BF16 or FP8 per-group quantized tensor. If it should be per-group quantized, to implement it efficiently in Triton, we reshape the output (BLOCK_M, BLOCK_N) to (BLOCK_M, BLOCK_N // G, G), where G is the group size. We then calculate the maximum value along the last dimension to get the scaling factor.

### G.3 TRITON-BASED FP8 KERNELS FOR NON-LINEAR LAYERS

1. Transposed FP8 Output: Besides generating the quantized FP8 output, these layers also produce a transposed FP8 output. This is critical for backward computations, where column-major inputs are required to calculate weight gradients.

2. Block Size Tuning: Block sizes for dimensions $M$ and $N$ are autotuned, with options chosen from $\{32, 64\}$.

### G.4 OPTIMIZER STATES

We implement a fused kernel for optimizer states update in CUDA. Since optimizer states use $1 \times 128$ quantization group size, every CUDA block calculate the result of a quantization group and contains 128 threads, each thread corresponds to one element. We uses warp shuffle functions for reduction to get the maximum absolute value and minimum absolute value within a quantization group to perform dynamic range expansion. We

Direct computation of $X_{\min}^k$ can result in underflow and numerical instability since its magnitude is usually about $10^{-12}$ and $k > 3$. To mitigate this issue, we introduce an additional FP32 value $C_X$, defined as $C_X = \sqrt{X_{\min} \cdot X_{\max}}$, to ensure numerical stability. By normalizing $X$ with $C_X$, the values $\frac{X_{\min}}{C_X}$ and $\frac{X_{\max}}{C_X}$ are computed, which are reciprocals of each other and closer to 1 compared to the original $X_{\min}^k$. This normalization enhances numerical robustness and reduces the risk of instability.

### G.5 OTHERS

1. Backward Gradient Handling: Backward gradients are computed in BF16 precision and are not quantized. This ensures that the gradient computations remain stable and accurate, avoiding numerical issues that can arise with quantization. Therefore for the backward kernels, the activation stored from forward is in FP8, while the input gradient and the output gradient are both in BF16 precision.

2. Gradient Accumulation Optimization: During the first gradient accumulation step, the scaling factors of weight matrices are stored. This avoids the need to recompute scaling factors in subsequent steps, reducing computational overhead. The FP8 weights are not stored to prevent additional memory usage, which is critical for large-scale models.

## H GROUP SIZE ANALYSIS

For activation, we calculate the quantization error of the output of last layer in forward pass, and the gradient before the first layer in backward pass, comparing with BF16. Based on Table below, we find that group size = 16 achieves a good balance for memory efficiency and quantization error.

| MSE | Only Quantize Linear | | Also Quantize Activation | | | | |
|---|---|---|---|---|---|---|---|
| | $1 \times 128$ Group | Per Tensor | 4 | 8 | 16 | 32 | 64 |
| Memory Overhead | - | - | 50% | 25% | 12.5% | 6.5% | 3.2% |
| Forward | 372 | 438 | 464 | 480 | 492 | 512 | 520 |
| Backward | 285 | 286 | 319 | 334 | 363 | 361 | 371 |

Table 10: Quantization overhead, forward, and backward MSE for different configurations and group sizes.

We also find that applying per-group quantization to linear layers is harmful for speed, while not improving the accuracy by too much. Even with system optimizations, this finer-grained method reduces the speed of linear layers by about 40% (when group size = 128, speed is reduced from 1300 TFlops to about 900 TFlops), greatly reducing the benefit of FP8. On the other hand, the accuracy gain is only marginal. Therefore we do apply per-tensor quantization for linear layers.

## I EFFICIENCY BREAKDOWN FOR OPTIMIZER AND ACTIVATION

We break down the improvement of each component and present the result in this table. Quantizing the optimizer states to FP8 is helpful in reducing the memory footprint, and can potentially increase the batch size to fully utilize the GPU. Quantizing the activation to FP8 can accelerate the training, and this number can be further improved by increasing the maximum batch size.

Table 11: Llama-2-7B Training Memory and Throughput Analysis

| Configuration | Optimizer | Activation | Peak | Max BS | Throughput | Speedup |
|---|---|---|---|---|---|---|
| **Llama-2-7B on 2 GPU** | | | | | | |
| BF16 | 26.2 GB | 25.8 GB | OOM | 1 | 6130 tokens/s | 1.00x |
| FP8 Optimizer | 6.4 GB | 25.8 GB | 61.9 GB | 2 | 7258 tokens/s | 1.18x |
| FP8 Optimizer + FP8 Activation | 6.4 GB | 16.9 GB | 35.6 GB | 4 | 11351 tokens/s | 1.85x |
| **Llama-2-7B on 4 GPU** | | | | | | |
| BF16 | 13.1 GB | 25.8 GB | 55.1 GB | 2 | 7730 tokens/s | 1.00x |
| FP8 Optimizer | 3.2 GB | 16.9 GB | 44.7 GB | 4 | 8384 tokens/s | 1.08x |
| FP8 Optimizer + FP8 | 3.2 GB | 16.9 GB | 35.6 GB | 4 | 11257 tokens/s | 1.45x |
| **Llama-2-7B on 8 GPU** | | | | | | |
| BF16 | 6.5 GB | 25.8 GB | 41.2 GB | 4 | 8238 tokens/s | 1.00x |
| FP8 Optimizer | 1.6 GB | 16.9 GB | 36.1 GB | 4 | 8444 tokens/s | 1.02x |
| FP8 Optimizer + FP8 Activation | 1.6 GB | 16.9 GB | 27.0 GB | 8 | 11241 tokens/s | 1.36x |

## J  OVERHEAD INTRODUCED BY GROUP SCALING

We show that the max reduction on each 1×G "can be seamlessly fused with the previous operation, adding minimal overhead". Take RMSNorm as an example, the speed of the RMSNorm kernel only increases by less than 5% when quantization is fused with it. Therefore, this adds only minimal overhead to the kernel.

Table 12: Performance Comparison: With and Without Quantization

| Matrix Size | With Quantize | Without Quantize |
|---|---|---|
| (4096, 1024) | 10.1 ms | 8.7 ms |
| (4096, 2048) | 29.6 ms | 28.7 ms |
| (4096, 4096) | 55.6 ms | 53.2 ms |
| (4096, 8192) | 122.6 ms | 117.5 ms |

## K  UNDERSTAND HOW EACH COMPONENT INFLUENCES THE MODEL PERFORMANCE

We separately quantize the optimizer and activations to better understand how each component influences the model performance. We find that quantizing activation incurs almost no degradation due to our well-designed quantization flow, and the main degradation comes from optimizer quantization. We choose to quantize both to maximize memory saving.

| | Train PPL | COPA | ARC (Easy) | SciQ | HellaSwag | Average |
|---|---|---|---|---|---|---|
| **BF16** | 2.995 | 60.0 | 45.6 | 67.3 | 33.7 | 51.6 |
| **FP8 O + A** | 3.008 | 61.0 | 44.2 | 67.6 | 33.7 | 51.5 |
| **FP8 O** | 2.998 (+0.003) | 60.0 (+0.0) | 44.5 | 66.0 | 34.1 | 51.1 |
| **FP8 A** | 2.999 (+0.004) | 63.0 (+3.0) | 44.0 | 68.4 | 34.1 | 52.3 |

Table 13: Performance comparison across different metrics and configurations.

## L  MORE RELATED WORKS

Due to page limit, some related works are presented in Appendix.

Table 14: Compared with DeepSeek-V3

| | | Non-Linear | | Attention | | | | Reduction Ratio | |
|---|---|---|---|---|---|---|---|---|---|
| | | RMSNorm | Act Func | RoPE | FlashAttn | Linear | Total | Ideal | Achieved |
| **Llama-style** | **BF16** | 4U | 8U | 2U | 3U | 5.66U | 22.66U | 1.00× | 1.00× |
| | **TE** | 2U | 8U | 2U | 3U | 3.33U | 18.33U | 1.23× | 1.20× |
| | **DeepSeek-V3** | 2U | 4U | 2U | 3U | 3.08U | 14.08U | 1.61× | - |
| | **COAT** | 1U | 4U | 2U | 3U | 3.33U | 13.33U | 1.69× | 1.65× |

**Activation Quantization** Recent research has focused on reducing memory footprint Cai et al. (2020) during neural network training through activation quantization. ActNN (Chen et al., 2021) introduced a 2-bit activation compressed training framework using random quantization, achieving 12x reduction in activation memory footprint1. GACT (Liu et al., 2022a) extended this concept to support various machine learning tasks and architectures, providing up to 8.1x reduction in activation memory for CNNs, transformers, and GNNs. However, these methods mostly focus on convolutional networks and are not applicable to LLMs. AQ-SGD (Wang et al., 2022) proposed compressing activation changes rather than direct values for pipeline-parallel training to reduce the communication overhead in the pipeline-parallel setting. Few-Bit Backward(Novikov et al., 2023) quantizes the non-linear activation functions using optimal piecewise-constant approximations to reduce the memory footprint while maintaining convergence. Jetfire (Xi et al., 2024) proposes INT8 data flow to quantize the activation of both linear and non-linear layers to reduce memory footprint and is applicable to language model pretraining.

**Bit-Width Under-Utilization** Balance Quantization Zhou et al. (2017) recursively partitions the parameters by percentiles to reduce quantization error. DDSQ Wang & Kang (2023) dynamically changes the quantization parameters according to different gradient distributions. N2UQ Liu et al. (2022b) improves the quantization precision via learning input thresholds and nonuniform mapping. These approaches are aware of the dynamic range problem in quantization and propose more adaptive, distribution-aware techniques that can preserve the nuanced information contained in full precision. They share a fundamental goal of expanding the effective dynamic range of low-precision neural network representations. These methods adopt a learnable non-uniform quantization lookup table, while our work relies on the natural FP8 data format.

