# OpenReview forum: "COAT: Compressing Optimizer states and Activations for Memory-Efficient FP8 Training"
_ICLR.cc/2025/Conference — ICLR 2025 Poster_

### Official Review · Reviewer_J45S · 2024-10-27

**Soundness:** 3
**Presentation:** 3
**Contribution:** 3
**Rating:** 6
**Confidence:** 4

**Summary:**

This work presents COAT, a new framework for FP8 training. There are two key designs. The first is a dynamic range expansion that tries to align the value range of optimizer states to that of FP8 representation, which improves the utilization of the FP8 value range and reduces quantization errors. The second is a mixed-granularity activation quantization, which combines (less nuanced) per-tensor quantization to linear layers and (more nuanced) per-group quantization to non-linear layers. Experimental results show that COAT achieves nearly the same convergence as BF16 training, yet is more time and memory efficient.

**Strengths:**

The paper is clearly written, well elaborating the problems of FP8 training as well as the proposed techniques to address the problems. Experiments on LLM pre-training, LLM fine-tuning, and VLM SFT demonstrate the sound convergence of COAT. And positive results present the improvement in terms of memory reduction and training speedup.

**Weaknesses:**

- The novelty of this work needs a justification.
- The ablation studies do not break down the improvement of each component.
- All discussions about FP8-LM are textual, lacking empirical comparisons in Section 6 and formal analysis in Table 2.

**Questions:**

(Q1) Dynamically scaling the value been has been proposed for long. For instance, almost all integer quantization approaches will scale the values with the max value or the norm of tensors. I suggest the authors discuss the novelty of their work, such as why exponential expand function is chosen.

(Q2) Per-group quantization can be done by first slicing the original tensor and then performing a per-slice quantization, which is implementation-specific. For me, a more important problem is how to determine the group size automatically. Currently there are no related discussions. And it is unclear how group sizes are determined in the experiments (e.g., based on hyper-parameter tuning or adaptive selection). If it is manually tuned, a sensitivity experiment would be nice to have.

Besides, in lines 83-85, the authors said that “per-tensor quantization for matrix multiplications is more efficient and better suited for TensorCores, while fine-grained quantization helps maintain accuracy.” I was expecting an empirical assessment on this, but failed to see any. Intuitively, fine-grained per-group quantization should always achieve lower quantization errors, then we should always prefer it to the per-tensor one if the efficiency gap is not significant.

(Q3) The convergence experiments are incredible. However, it would be better to break down the improvement of each design in your work. In addition, breaking down the speedup and memory reduction for different modules in the model can help readers better understand the effectiveness.

(Q4) FP8-LM, one of the most relevant works, is only briefly discussed, while not included in the analysis and experiments in the main body.

(Q5) In lines 333-334, the authors said that the max reduction on each $1 \times G$ “can be seamlessly fused with the previous operation, adding minimal overhead”. Can you provide more details about how this is achieved?

(Q6) Minor comments:
- Line 192: “can not” should be “cannot”.
- Footnote 2: “RoPE and FlashAttention is …” should be plural.
- In Table 2 and Appendix, it might be more appropriate to set 1U to be FP8, rather than BF16.

---

> ### Author Response · Authors · 2024-11-23
>
> We thank the reviewer for acknowledging our novel contributions and insightful questions. Below we respond to the questions. We will revise the paper's final versions based on the reviewers' comments.
>
> ### **Q1**: Justification of the novelty of this work
>
> **A1**: COAT's key contributions include:
>
> Optimizer Quantization:
>  - COAT **first identifies the dynamic range under-utilization problem** for optimizer state quantization, which is crucial for optimizer states quantization. We proposes the dynamic range expansion method to address the problem, and prove that our polynomial expand function is the **theoretically optimal** expand function. We also empirically verify that our choice can significantly reduce the MSE and do not lead to performance degradation.
>
> Activation Quantization:
>  - Propose to use distinct quantization policies for linear and non-linear layers to best preserve accuracy and optimize memory efficiency, addressing gaps left by previous work that primarily focused on compute efficiency. Previous FP8 training works only focus on compute efficiency and overlook the importance of memory efficiency. While such insights have been extensively studied in inference, they are rarely explored in training due to the complexity of the training workflow and error accumulation.
>
> COAT’s innovations address critical challenges in FP8 quantization and system optimization. COAT democratizes LLM training, reducing GPU requirements while maintaining high throughput, making it highly practical for real-world scenarios.
>
> ### **Q2**: Break down the improvement of each component in ablation study.
>
> **A2**: Thanks for the constructive advice! We present a more detailed analysis for Table 7 to do an ablation study on the improvement of each component. We conduct experiments on Llama-2-7B and vary the number of devices from 2, 4, and 8.
>
> We find that quantizing the optimizer states to FP8 is more helpful when the number of GPU <= 4, and quantizing the activation to FP8 is more beneficial when the number of GPU >= 8.
>
> | Llama-2-7B on 2 GPUs           | Optimizer Memory | Activation Memory | Peak Memory | Maximum Training Throughput | Speedup Ratio |
> |--------------------------------|------------------|-------------------|-------------|-----------------------------|---------------|
> | BF16 | 26.2 GB  | 25.8 GB | OOM         | BS=1, 6130 token/s            | 1.00x         |
> | FP8 Optimizer  | 6.4 GB  | 25.8 GB           | 61.9 GB     | BS=2, 7258 token/s           | 1.18x         |
> | FP8 Optimizer + FP8 Activation | 6.4 GB           | 16.9 GB           | 35.6 GB     | BS=4, 11351 token/s           | 1.85x         |
> |   |      |       |    |     |     |
> | Llama-2-7Bon 4 GPUs            | Optimizer Memory | Activation Memory | Peak Memory | Maximum Training Throughput | Speedup Ratio |
> | BF16                           | 13.1 GB          | 25.8 GB           | 55.1 GB     | BS=2, 7730 token/s            | 1.00x         |
> | FP8 Optimizer                  | 3.2 GB           | 16.9 GB           | 44.7 GB     | BS=4, 8384 token/s           | 1.08x         |
> | FP8 Optimizer + FP8 Activation | 3.2 GB           | 16.9 GB           | 35.6 GB     | BS=4, 11257 token/s           | 1.45x         |
> |                                |                  |                   |             |                             |               |
> | Llama-2-7Bon 8 GPUs            | Optimizer Memory | Activation Memory | Peak Memory | Maximum Training Throughput | Speedup Ratio |
> | BF16                           | 6.5 GB           | 25.8 GB           | 41.2 GB     | BS=4, 8238 token/s           | 1.00x         |
> | FP8 Optimizer                  | 1.6 GB           | 16.9 GB           | 36.1 GB     | BS=4, 8444 token/s           | 1.02x         |
> | FP8 Optimizer + FP8 Activation | 1.6 GB           | 16.9 GB           | 27.0 GB     | BS=8, 11241 token/s          | 1.36x         |
>
> ### **Q3**: Empirical comparisons about FP8-LM in Section 6 and Table 2.
>
> **A3**: We would like to clarify that **COAT is orthogonal to FP8-LM**. FP8-LM mainly quantizes the (1) **gradients** to FP8 to reduce communication overhead, (2) **parameters** to FP16 to save memory, and (3) first-order momentum to FP8 to save memory. In contrast, we mainly focus on quantizing the **first-order and second-order optimizer states** and **activations** to FP8.
>
> We updated Table 2 and observed that FP8-LM has an equivalent effect on activation memory as TE. As shown in Table 2, COAT delivers 1.51x higher activation memory saving over FP8-LM. It is possible to combine COAT with FP8-LM to boost training efficiency further. However, it is out of the scope of this work and we leave it to future research.

---

> ### Author Response · Authors · 2024-11-23
>
> ### **Q4**: Dynamically scaling the value has been proposed for a long. For instance, almost all integer quantization approaches will scale the values with the max value or the norm of tensors. I suggest the authors discuss the novelty of their work, such as why the exponential expansion function is chosen.
>
> **A4**: To our best knowledge, the range underutilization problem of optimizer state quantization is first identified in COAT [1] [2]. This under-utilization problem **cannot be resolved** by traditional quantization methods, which typically scale tensors by dividing them with a scale factor based on their maximum value. While this normalizes the tensor, it does not alter the ratio between the maximum and minimum values (i.e., the dynamic range), leading to significant quantization errors.  We also **empirically verify** that this method decreases the MSE of m / \sqrt{v} by 1.63x, as shown in Table 1 in our paper.
>
> |                             | w/o Dynamic Range Expansion | w/ Dynamic Range Expansion |
> |-----------------------------|-----------------------------|----------------------------|
> | MSE of $\frac{m}{\sqrt{v}}$ | 20.10                       | 12.31                      |
>
> For the expansion function chosen by COAT, We **mathematically prove** that our polynomial expand function $f(x) = x^k$ is **optimal** and attach the proof in Appendix G. Importantly, if the expansion function is not chosen carefully, the MSE error will become suboptimal.
>
> | MSE         | No expansion | Polynomial (Ours) | Exponential |
> |-------------|--------------|-------------------|-------------|
> | E4M3 + E4M3 | 22.28        | 12.07             | 27.99       |
> | E4M3 + E5M2 | 28.34        | 11.94             | 21.68       |
> | E5M2 + E4M3 | 40.81        | 14.66             | 20.02       |
> | E5M2 + E5M2 | 44.41        | 14.55             | 12.59       |
>
> [1] Dettmers, Tim, et al. "8-bit optimizers via block-wise quantization." arXiv preprint arXiv:2110.02861 (2021).
>
> [2] Li, Bingrui, Jianfei Chen, and Jun Zhu. "Memory efficient optimizers with 4-bit states." Advances in Neural Information Processing Systems 36 (2024).
>
> ### **Q5**: How group sizes are determined in the experiments
>
> **A5**: We choose to set the group size = 16 for activation quantization. This choice is based on **balancing the memory efficiency overhead and the quantization error**. With a smaller group size, the quantization error is reduced but will introduce high memory overhead on the scaling factors (bfloat16).
>
> We calculate the quantization error of the output of the last transformer layer in the forward pass, and the gradient before the first transformer layer in the backward pass, compared with BF16. Based on the Table below, we find that group size = 16 achieves a good balance for memory efficiency and quantization error. We include these results in Appendix H to improve our paper.
>
> | MSE      | Linear use 1 * 128 quant | Only Quantize Linear | 4   | 8   | 16    | 32   | 64   |
> |----------|--------------------------|----------------------|-----|-----|-------|------|------|
> | Overhead | -                        | -                    | 50% | 25% | 12.5% | 6.5% | 3.2% |
> | Forward  | 372                      | 438                  | 464 | 480 | 492   | 512  | 520  |
> | Backward | 285                      | 286                  | 319 | 334 | 363   | 361  | 371  |
>
> ### **Q6**: Compare per-tensor quantization and per-group quantization for matrix multiplications
> We find that per-group quantization is not suitable for matrix multiplication since it makes **linear layers much slower because of frequent quantized and dequantize operations**, and the **accuracy gain is not significant**. This phenomenon is also observed in Qserve[1].
>
> Meanwhile, using very fine-grained quantization for linear layers does not bring much benefit compared with per-tensor quantization in terms of MSE, as reported in the table above in Q5. Therefore we do not apply per-group quantization to linear layers.
>
> [1] Lin, Yujun, et al. "Qserve: W4a8kv4 quantization and system co-design for efficient llm serving." arXiv preprint arXiv:2405.04532 (2024).

---

> ### Author Response · Authors · 2024-11-23
>
> ### **Q7**: Details about the max reduction on each 1×G group in our Group Scaling design “can be seamlessly fused with the previous operation, adding minimal overhead”
>
> **A7**: We take the RMSNorm + Linear as an example. Input size is (M, N), group size is G. In our method, the RMSNorm kernel has 6 parts:
> 1. Load the FP8 input of size (1, N) and BF16 scaling factor of size (1, N // G) to chip
> 2. Dequantize the input to float32
> 3. Calculate the RMSNorm result
> 4. **Calculate the maximum value for every G value, to get a tensor of shape (1, N // G).**
> 5. Store the output of step 3 to HBM.
> 6. **Store the maximum value in step 4 to HBM**
>
> The overhead we introduce is step 4 and step 6. If we fuse these 2 steps, the latency of the RMSNorm kernel only increases by less than 5%.
>
> |              | With Quantize (step 1,2,3,4,5,6) | Without Quantize (step 1,2,3,5) |
> |--------------|----------------------------------|---------------------------------|
> | (4096, 1024) | 10.1ms                           | 8.7ms                           |
> | (4096, 2048) | 29.6ms                           | 28.7ms                          |
> | (4096, 4096) | 55.6ms                           | 53.2ms                          |
> | (4096, 8192) | 122.6ms                          | 117.5ms                         |
>
> ### **Q8**: Minor comments:
>
> **A7**: Thanks for the suggestion and we have revised accordingly.

---

> > ### Comment · Reviewer_J45S · 2024-11-25
> >
> > Thanks for the rebuttal along with the additonal experimental results, which address my concerns.

---

> > > ### Author Response · Authors · 2024-11-25
> > >
> > > Thanks for your time and effort. We are glad that our response addressed your concerns. Could you please consider improving the rating? If you have any more comments, we are always willing to provide clarifications.

---

> ### Author Response · Authors · 2024-11-25
>
> Dear Reviewer J45S,
>
> We are looking forward to further communication and please let us know if our response have fully addressed your concerns.
>
> Sincerely,
>
> Authors

---

### Official Review · Reviewer_gVQo · 2024-11-08

**Soundness:** 3
**Presentation:** 3
**Contribution:** 2
**Rating:** 6
**Confidence:** 3

**Summary:**

This paper introduces COAT, a scheme for efficient low-precision training of large models. It is based on compressing optimizer states and model activation through two mechanisms: _(i) Dynamical Range Expansion_ for fully exploiting the representation range of the FP8 format for the optimizer states) and _(ii) Mixed-Granularity Activation Quantization_, which uses a combination of per-tensor and per-group quantization strategies to compress model's activations.

Extensive experiments are conducted on LLMs (OLMo-1/7B, LLaMa 2/7B) and VLMs (VILA1.5-7B), showing solid and consistent improvements in (i) memory efficiency and (ii) training speed, at nearly lossless performance.

**Strengths:**

In my opinion, this paper has the following main strenghts:
- **Clear presentation:** the paper is very clearly written, both the introduction and the related work sections introduce the necessary notions the proposed approach is built on and improves upon. All the figures/tables are of good quality and are well displayed in a printed version of the manuscript.
- **Practical significance:** as highlighted by the authors, efficient training of large models is a timely problem, the ideas presented in this work are relatively simple but proven effective in realistic scenarios. Overall, I believe that the proposed approach has the potential to be integrated in classical schemes for large-scale training.

**Weaknesses:**

The main points of discussion of this submission are in my opinion related to:
- **Novelty/Fit for ICLR:** the proposed techniques are not strictly novel in terms of research. Indeed, while the approach itself is technically new (e.g. at the best of my knowledge has not been proposed in previous works) it uses a combination of state-of-art techniques, with a strong engineering focus on how to integrate them in the most effective way. For example, the proposed _Dynamic Range Expansion_ is a simple expansion of the domain values of the optimizer states to the range of the used representation, and the _Mixed-Granularity Activation Quantization_ uses VS-Quant and Per-Block Quant for non-linear layers and per-tensor quantization for linear layers, which are not themselves new techniques.
- **Missing implementation details:** given that the work is purely experimental and that reproducing the results strongly relies on efficient implementation, one would expect more implementation details. The provided details are sufficient for an high-level understanding of the proposed approach but still require significant effort to reproduce the results. Ideally, one would like to have a working code attached to the submission, such that reviewers can try it. Indeed, at the current state, it is not possible for a reviewer to reproduce the results.

In essence, I believe that this paper has mainly (good) incremental engineering contributions, while I have some concerns about its contributions in term of research ideas. Overall, I do question if it is a good fit for ICLR, rather than the validity of the approach or its practical usefulness.
I provisionally give a "Borderline accept" and wait for the next phase to discuss with other reviewers.

**Questions:**

- Can you discuss better the approaches related to Activation Quantization? From lines 128-132 is not clear what are the improvements and the novelty of the proposed approach with respect to previous works.
- Can you provide a deeper explanation of what are the main differences between the proposed Group Scaling and the TransformerEngine's Delayed Scaling? In the text it is mentioned that it causes instabilities, but it is not clear what they are and why they arise.

Small issues:
- There is a missing ref. in line 372: "performance in [] 4".
- Table 6 is a little bit confusing in reporting the memory saving and the speedup, e.g. it is not immediately clear that the first "Ratio" column refers to runtime and the second to memory usage. I suggest to make it clearer, for example by adding a vertical line after the first "Ratio" column.

---

> ### Author Response · Authors · 2024-11-23
>
> We thank the reviewer for acknowledging our novel contributions and insightful questions. Below we respond to the questions. We will revise the paper's final versions based on the reviewers' comments.
>
> ### **Q1**: Novelty/Fit for ICLR
>
> **A1**: From ICLR 2025 Call for papers [1], one important relevant topic of ICLR is
>
>  - “infrastructure, software libraries, hardware, etc” and “considers a broad range of subject areas including ……. **applications** in vision, audio, speech, language …… in ML.”
>
> COAT proposes a novel FP8 training framework for **large-scale model training** and democratizes the language models, thus undoubtedly fit the scope of ICLR conference. In past acceptance of ICLR,  awesome projects such as LBA Inference [2], ZeRO++ [3], and LUQ [4] have demonstrated the importance of extensive system optimizations in identifying the best practices for real-world scenarios. The insights brought to the community are equally important as the techniques used.
>
> Further, COAT delivers significant value to the **LLM community** by enabling FP8 quantization to be effectively used in real-world training scenarios. Its contributions directly address the **scalability and cost-efficiency challenges** faced by researchers and practitioners by allowing large language models (LLMs) to train effectively on setups with fewer GPUs while also having a high training throughput.
>
> ### **Q2**: combination of state-of-art techniques, with a strong engineering focus, xxx. which are not themselves new techniques
>
> **A2**: We arguably disagree with the comments and clarify as follows:
>
> The dynamic range expansion is motivated by a clear rationale, addressing the under-utilization of the dynamic range in optimizer states. Without this expansion, significant quantization errors can occur. If the expand function is not chosen carefully, it may also result in sub-optimal results.
>
> | MSE         | No expansion | Polynomial (Ours) | Exponential |
> |-------------|--------------|-------------------|-------------|
> | E4M3 + E4M3 | 22.28        | 12.07             | 27.99       |
> | E4M3 + E5M2 | 28.34        | 11.94             | 21.68       |
> | E5M2 + E4M3 | 40.81        | 14.66             | 20.02       |
> | E5M2 + E5M2 | 44.41        | 14.55             | 12.59       |
>
> In COAT, the polynomial expansion comes from a theoretical formula and we have proved its optimal for quantization optimizer states (details in Appendix F). To our best knowledge, the formula, as well as the range under-utilization is first discussed for optimizer states.
>
> Applying different quantization policies for different layers is rarely discussed and most previous studies have focused on the inference. Existing solutions such as TransformerEngine[5], FP8-LM[6], and FP8 to Trillion Token[7]  only use per-tensor quantization for linear layers and do not quantize non-linear layers to best preserve accuracy. COAT makes an important step to transform most dataflows into FP8 for better efficiency.
>
> Therefore, we firmly believe that COAT will also make a valuable contribution to the conference and inspire future low-bit training researchers.
>
> [1] https://iclr.cc/Conferences/2025/CallForPapers
>
> [2] Towards Cheaper Inference in Deep Networks with Lower Bit-Width Accumulators https://openreview.net/forum?id=oOwDQl8haC
>
> [3] ZeRO++: Extremely Efficient Collective Communication for Large Model Training https://openreview.net/forum?id=gx2BT0a9MQ
>
> [4] Accurate Neural Training with 4-bit Matrix Multiplications at Standard Formats: https://openreview.net/forum?id=yTbNYYcopd
>
> [5] https://github.com/NVIDIA/TransformerEngine
>
> [6] Peng, Houwen, et al. "Fp8-lm: Training fp8 large language models." arXiv preprint arXiv:2310.18313 (2023).
>
> [7] Fishman, Maxim, et al. "Scaling FP8 training to trillion-token LLMs." arXiv preprint arXiv:2409.12517 (2024).

---

> ### Author Response · Authors · 2024-11-23
>
> ### **Q3**: More discussion about implementation details
>
> **A3**: We have uploaded our codebase to anonymous GitHub for preview: https://anonymous.4open.science/r/COAT-Anonymous-E13B/ and will open source when less anonymous. The COAT LLaMA implementation introduces three FP8-optimized modules: BeforeAttention, AfterAttention, and MLPResidual. These modules, built using torch.autograd.Function, ensuring accurate gradient flow computation. LayerNorm input quantization is fused with the residual connection to boost performance.
>
> Custom FP8 matrix multiplication kernels for linear layers are implemented with Triton and optimized using triton.autotune, with M, N, and K dimensions tuned from {128, 256} based on hardware. Group scaling reshapes outputs to (BLOCK_M, BLOCK_N//G, G) before linear layers, computes scaling factors for each group, and stores them efficiently.
>
> Non-linear layers also leverage Triton-based FP8 kernels. The forward pass generates quantized FP8 and transposed FP8 outputs, critical for efficient backward computations requiring column-major inputs. This avoids costly reordering during backpropagation, enhancing performance. Block sizes {32, 64} are autotuned to ensure optimal performance across all FP8 computations.
>
> To replicate our result using our anonymous code, you can first install coat and the fp8 optimizer kernel. Then download the dataset of OLMo.
>
> To reproduce the memory reduction result, run:
>
> BF16 baseline:
>
> MEMORY_BENCH=1 torchrun --nproc_per_node=4 scripts/train.py configs/reproduce/OLMo-7B-reproduce-MemBench.yaml
>
> COAT:
>
> MEMORY_BENCH=1 torchrun --nproc_per_node=4 scripts/train.py configs/coat/OLMo-7B-COAT-Both-MemBench.yaml
>
> To reproduce the memory reduction result, run the following:
>
> BF16:
>
> SPEED_BENCH=1 torchrun --nproc_per_node=4 scripts/train.py configs/reproduce/OLMo-7B-reproduce-SpeedBench.yaml
>
> COAT:
>
> SPEED_BENCH=1 torchrun --nproc_per_node=4 scripts/train.py configs/coat/OLMo-7B-COAT-Both-SpeedBench.yaml
>
> To reproduce the OLMo pretraining experiment, run:
> torchrun --nproc_per_node=8 scripts/train.py configs/coat/OLMo-1B-COAT-BOTH.yaml
>
> ### **Q4**: Difference with other related works about activation quantization?
>
> **A4**: Recent activation quantization approaches collectively focus on the memory saving but ignore the compute efficiency, and most of them are designed for neural networks and not optimized for recent large language models. COAT’s design the activation quantization with FP8 precision flow, thus enjoying both memory- and compute-efficiency. COAT also extensively verifies its accuracy on LLM related tasks.
>
> For example, ActNN[1] introduced a 2-bit activation compressed training framework using random quantization, achieving a 12x reduction in activation memory footprint1. GACT[2] extended this concept to support various machine learning tasks and architectures, providing up to 8.1x reduction in activation memory. However, these methods mostly focus on convolutional networks and are not applicable to LLMs. AQ-SGD[3] proposed compressing activation changes rather than direct values for pipeline-parallel training to reduce the communication overhead. Few-Bit Backward[4] quantizes the non-linear activation functions using optimal piecewise-constant approximations to reduce the memory footprint of activation functions.. Jetfire[5] proposes INT8 data flow to quantize the activation of both linear and non-linear layers to reduce memory footprint and is applicable to language model pretraining. We modify lines 127-138 to discuss these related works in detail.
>
> [1] Chen, Jianfei, et al. "Actnn: Reducing training memory footprint via 2-bit activation compressed training." International Conference on Machine Learning. PMLR, 2021.
> [2] Liu, Xiaoxuan, et al. "Gact: Activation compressed training for generic network architectures." International Conference on Machine Learning. PMLR, 2022.
> [3] Wang, Jue, et al. "Fine-tuning language models over slow networks using activation quantization with guarantees." Advances in Neural Information Processing Systems 35 (2022): 19215-19230.
> [4] Novikov, Georgii Sergeevich, et al. "Few-bit backward: Quantized gradients of activation functions for memory footprint reduction." International Conference on Machine Learning. PMLR, 2023.
> [5] Xi, Haocheng, et al. "Jetfire: Efficient and accurate transformer pretraining with int8 data flow and per-block quantization." arXiv preprint arXiv:2403.12422 (2024).

---

> ### Author Response · Authors · 2024-11-23
>
> ### **Q5**: Main differences between the proposed Group Scaling and the TransformerEngine's Delayed Scaling
>
> **A5**: Delay scaling **estimates the scaling factor of recent activations history**, while **our group scaling uses the current step’s activation to calculate the scaling factor**. Our group scaling and delay scaling both reduce the latency when calculating the scaling factor.
>
> In a linear layer, suppose we train for $N$ steps, with activations $X_1, \dots, X_N$. When the activation $X_{N+1}$ arrives, delay scaling calculates the scaling factor for $X_{N+1}$ by recording the maximum value from a sliding window of recent activations, $X_{N-w}, \dots, X_N$, where $w$ is the window size. Our group scaling will calculate the scaling factor based on the maximum value of  $X_{N+1}$.
>
> We propose group scaling instead of delayed scaling because of stability issues, which will expand below.
>
> ### **Q6**: Delayed Scaling causes instabilities, but it is not clear what they are and why they arise.
>
> **A6**: In pretraining scenarios, **large activation spikes** can produce unusually high activation values. If scaling factors are determined solely based on historical statistics, they may be insufficient to accommodate the larger values of XN+1X_{N+1}, which demand a significantly higher scaling factor. This mismatch results in excessive quantization during the current step, causing substantial quantization errors in the training process.
>
> In contrast, our method uses the statistics of $X_{N+1}$ to calculate an appropriate scaling factor, ensuring accurate quantization and mitigating the impact of these activation spikes.

---

> ### Author Response · Authors · 2024-11-25
>
> Dear Reviewer gVQo,
>
> We are looking forward to further communication and please let us know if our response have fully addressed your concerns.
>
> Sincerely,
>
> Authors

---

> ### Author Response · Authors · 2024-12-02
>
> Dear Reviewer gVQo,
>
> We truly appreciate your time and effort in reviewing our submission and offering valuable suggestions. The author-reviewer discussion period is closing soon, so we would like to confirm whether our responses have effectively addressed your questions and concerns.
>
> Thank you for your time and insights.
>
> Best regards,
>
> Authors

---

> ### Author Response · Authors · 2024-12-03
>
> Dear Reviewer gVQo,
>
> Thank you again for your detailed suggestions and comments. As the deadline of discussion period is approaching, could you kindly review our responses and inform us whether you have further questions?
>
> Thank you!
>
> Sincerely,
>
> The Authors

---

### Official Review · Reviewer_P9H9 · 2024-11-08

**Soundness:** 3
**Presentation:** 3
**Contribution:** 2
**Rating:** 6
**Confidence:** 3

**Summary:**

This work proposes a quantization improvement for transformers that mainly targets the optimizer’s states and activations. The authors propose a FP8 per-group quantization of the optimizer’s states along with a “dynamic range expansion” that utilizes the quantized space more effectively. The FP8 per-group quantization is also applied on the activations with "mixed-granularity", i.e., different layers quantize their inputs differently (e.g., layernorm vs. linear). These improvements can save up to 1.5x memory over TransformerEngine and is slightly faster as well without sacrificing performance vs. FP16 quantization.

**Strengths:**

- The experiments demonstrate significant space improvements in practice and slight speed up.
- Figures illustrates methodology and motivation clearly.
- The authors motivate the quantization method by looking at the low-bit values as a range of representable values, and show that optimizer states are sparse in this representation and do not span the range nicely. A transformation $f(x)=x^k$ can be applied before quantization and inverted after dequantization easily, allowing for the quanitized values to span the range of representable values more uniformly.
- Table 2 shows the actual improvements in memory clearly.
- The mixed-granularity approach is simple yet effective in practice.

**Weaknesses:**

The scientific contribution of this work is limited. Despite the practical improvements, it might be of little interest to the community of “learning representations”, but perhaps of great interest to LLM practitioners, which is certainly a significant part of the AI community. Thus, I will not wholly judge the paper by its scientific merit and shall take a pragmatic approach in my rating here, but the point regarding the contribution still stands.

The authors could have, for example, expanded on Sec 4.1. They could have studied or tried other inveritble transformations that spread out the values nicely across the representable values. Would direct training in this transformed low-bit regime work? Perhaps the author can study whether polynomial activations can have the same benefits, for example. This would make the paper more interesting.

Another thing is that the idea behind the dynamic range is quite straightfoward to conceive, so I believe a more careful literature review can reveal very similar techniques. For example, a quick search reveals [1], which studies a similar problem.

Finally, the engineering improvements to transformer quantization are indeed non-trivial but can be found by extensive experimentations and ablation studies, which might not be possible for those who cannot afford to run such extensive experiments. Thus, I believe more effort should go into justifying the engineering choices found by the authors, which would be greatly appreciated and would motivate similar practices in the future. For example, a more fine-grained ablation studiy or theoretical analysis of the mixed-grained quantization can be great. This should also inform future readers on whether this particular quantization method is the best one for some particular setting.

[1] Balanced Quantization: An Effective and Efficient Approach to Quantized Neural Networks. Zhou et al. 2017.

**Questions:**

None.

---

> ### Author Response · Authors · 2024-11-23
>
> We thank the reviewer for acknowledging our contributions and the insightful comments. We respond to the questions below. We will revise the paper's final versions based on the reviewers' comments.
>
> ### **Q1**: Clarify the scientific contribution of this work:
>
> **A1**: While we appreciate the reviewers recognizing the “scientific metric/pragmatic approach” of COAT, we would like to politely yet firmly emphasize the algorithm-wise contribution of COAT:
>
>  - Dynamic range expansion that **firstly identifies** the range under-utilization issue and designs **theoretical optimal** expansion function (General response for details).
>
>  - Different quantization policies for linear/non-linear layers **best mitigates the accuracy gap** and **maximize the memory efficiency** while previous work only focuses on compute efficiency.
>
> COAT has not only theoretically proven the optimal expansion function, but also extensively verified the performance on a wide range of tasks, as well as the choice of quantization strategy for different layers. We hope that reviewers can provide a holistic view of COAT’s scientific contribution.
>
> ### **Q2**: Little interest in the community of “learning representations”,
>
> **A2**: COAT as an efficient training technique is a **good fit for the ICLR community**. While the ICLR community initiates from the interests of “learning representations”,  From ICLR 2025 Call for papers [1], one important relevant topic of ICLR is
>
>  - “infrastructure, software libraries, hardware, etc” and “considers a broad range of subject areas including ……. applications in vision, audio, speech, language …… in ML.”
>
> In the past acceptance of ICLR, awesome projects such as LBA Inference [2], ZeRO++ [3], and LUQ [4] have demonstrated the importance of extensive system optimizations in identifying the best practices for real-world scenarios. COAT proposes a novel FP8 training framework for large-scale model training and democratizes the language models, thus surely fitting the scope of the ICLR conference.
>
> [1] https://iclr.cc/Conferences/2025/CallForPapers
>
> [2] Towards Cheaper Inference in Deep Networks with Lower Bit-Width Accumulators. https://openreview.net/forum?id=oOwDQl8haC
>
> [3] ZeRO++: Extremely Efficient Collective Communication for Large Model Training. https://openreview.net/forum?id=gx2BT0a9MQ
>
> [4] Accurate Neural Training with 4-bit Matrix Multiplications at Standard Formats. https://openreview.net/forum?id=yTbNYYcopd
>
> ### **Q3**: studied or tried other invertible transformations that spread out the values nicely across the representable values.
>
> **A3**: Thanks for the insightful question. In fact, the transformation (expansion function) has to be chosen carefully. In COAT, We **theoretically prove** that the choice of $f(x) = x^k$ is optimal for the expansion function (details in newly added Appendix F). Other invertible transformations are **suboptimal** in theory and lead to larger MSE errors as we will show below.
>
> Following the suggestion, we also test other invertible transformations. We choose the $\frac{e^{kx} - 1}{e^k - 1}$ function and use $k$ to control the strength. To avoid numerical problems, we divide $x$ by its maximum value before expansion, therefore the input is restricted to (0, 1]. We find that the quantization error of the exponential function underperforms our polynomial expansion function.
>
> | MSE         | No expansion | Polynomial (Ours) | Exponential |
> |-------------|--------------|-------------------|-------------|
> | E4M3 + E4M3 | 22.28        | 12.07             | 27.99       |
> | E4M3 + E5M2 | 28.34        | **11.94**             | 21.68       |
> | E5M2 + E4M3 | 40.81        | 14.66             | 20.02       |
> | E5M2 + E5M2 | 44.41        | 14.55             | **12.59**       |
>
> ### **Q4**: Would direct training in this transformed low-bit regime work?
>
> **A4**: We find that direct training when representing optimizer states with FP8 does not work for training, and the loss curve will quickly diverge after even 100 steps. The scaling factor and the expand function are crucial to stabilize the training process.

---

> ### Author Response · Authors · 2024-11-23
>
> ### **Q5**: Whether polynomial activations can have the same benefits, for example.
>
> **A5**: Applying dynamic range expansion to activation will **significantly slow the training, but the accuracy gain is relatively marginal**. We report the latency of the quantization operator, which demonstrates a nearly threefold increase in quantization overhead. And even if we apply dynamic range expansion to activation, the quantization error is not reduced significantly. Consequently, we opted not to adopt the dynamic range expansion technique to activations in our method.
>
> | Matrix Size | Regular (ms) | Expand (ms) |
> |-------------|--------------|-------------|
> | 1024 * 1024 | 0.05         | 0.13        |
> | 2048 * 2048 | 0.11         | 0.26        |
> | 4096 * 4096 | 0.21         | 0.60        |
> | 8192 * 8192 | 0.41         | 1.24        |
>
> ### **Q6**: Justifying the engineering choices, such as fine-grained ablation study or theoretical analysis of the mixed-grained quantization
>
> **A6**: To **balance overhead and accuracy**, we set the group size to 16. We explored group sizes ranging from 4 to 64 to compare their respective overheads and quantization errors, as shown in the table below.
>
> When the group size of FP8 quantization is 16, each quantization group requires 8×16+16=144 bits in total, resulting in 9 bits per element. This introduces a 12.5% memory overhead compared to the ideal 8-bit-per-element scenario. The quantization error is measured using the Mean Squared Error (MSE) of the output values in the last layer between BF16 and FP8 training.
>
> Smaller group sizes (e.g., 4) yield lower quantization errors but incur higher overhead. Larger group sizes (e.g., 64) reduce overhead but increase forward or backward quantization errors. Based on these statistics, we chose to set the group size to 16.
>
> | MSE      | Only Quantize Linear | 4   | 8   | 16    | 32   | 64   |
> |----------|----------------------|-----|-----|-------|------|------|
> | Overhead | -                    | 50% | 25% | 12.5% | 6.5% | 3.2% |
> | Forward  | 438                  | 464 | 480 | 492   | 512  | 520  |
> | Backward | 286                  | 319 | 334 | 363   | 361  | 371  |
>
> We add the related discussions to Appendix H to improve our paper.
>
> ### **Q7**: More careful literature review for dynamic range under-utilization problem
>
> **A7**: Thanks for the suggestions.  We have added more literature [1-3] in lines 141-148, Section 2 in our updated paper and will further revise it in our final version.
>
> [1] Zhou, Shu-Chang, et al. "Balanced quantization: An effective and efficient approach to quantized neural networks." Journal of Computer Science and Technology 32 (2017): 667-682.
>
> [2] Wang, Shuai, and Yi Kang. "Gradient distribution-aware INT8 training for neural networks." Neurocomputing 541 (2023): 126269.
>
> [3] Liu, Zechun, et al. "Nonuniform-to-uniform quantization: Towards accurate quantization via generalized straight-through estimation." Proceedings of the IEEE/CVF conference on computer vision and pattern recognition. 2022.

---

> > ### Comment · Reviewer_P9H9 · 2024-11-23
> >
> > Thank you for the rebuttal and for following up with experimental results. Also see my comment above regarding the proof. Overall, I am satisfied with the response.
> >
> > Regarding the scientific merit, I hope that the authors do not take my comment as a criticism. It is just a (slightly biased) comment regarding the nature of this work, which is well-suited for ICLR as the authors pointed out.

---

> > > ### Author Response · Authors · 2024-11-25
> > >
> > > Thank you for your kind words and timely feedback—especially on a weekend! We appreciate your time and effort in helping improve our work!
> > >
> > > Could you consider improving the rating if our response has resolved all of your concerns and you think COAT is a good fit for ICLR? If not, please let us know if there is any additional experiments / proofs we can make to better clarify.

---

> > > > ### Comment · Reviewer_P9H9 · 2024-11-25
> > > >
> > > > Dear Authors, thank you for your eagerness to provide clarifications and improvements. I believe the current rating reflects my opinon about this work more accurately, so I would like to keep it.

---

### Official Review · Reviewer_4rZz · 2024-11-11

**Soundness:** 3
**Presentation:** 3
**Contribution:** 2
**Rating:** 6
**Confidence:** 2

**Summary:**

This paper introduces a novel fp8 training framework named COAT that improves throughput and memory footprint of (end-to-end) large-scale training respectively by 1.54x and 1.43x. In detail, COAT is built upon two key techniques of (1) dynamic range expansion that allows for abetter utilization of fp8 format’s representation range, and (2) mixed-granularity activation quantization that reduces the memory footprint for activations. The authors demonstrate that COAT achieves superior memory/compute efficiency while preserving the bf16 performance on three tasks of LLM pretraining, fine-tuning, and VLM training.

**Strengths:**

1. Given that H100 is a fairly new hardware, fp8 training has not been explored extensively in previous literature. At the same time, the potential memory/compute efficiency gain from fp8 training can be significant, so I believe this paper addresses an important problem and makes a progress in this topic.
2. Dynamic range expansion is well-motivated by the analysis from Figure 2.
3. I believe ~1.5x improvements in memory/compute efficiency is significant.

**Weaknesses:**

1. Even if the performance gap looks not too significant, COAT still underperforms bf16 training on most tasks. At the same time, the ablation study only focuses on memory/compute efficiency, but not the final model performance. Providing insights on how each component influences the model performance and insights on the source of the performance degradation can be useful.
2. It is unclear to me how/whether the dynamic range is synchronized across GPUs in the distributed training setting.

**Questions:**

See above.

---

> ### Author Response · Authors · 2024-11-23
>
> Thanks for your valuable comments and feedback. We respond to the questions below.
>
> ### **Q1**: COAT still underperforms bf16 training on most tasks
>
> **A1**: COAT achieves on-par performance with BF16, and we would like to clarify the misunderstanding regarding COAT’s performance. For LLM results, 3 out of 8 cases outperform BF16 by an average of 1.0%, 3 out of 8 underperform by an average of 1.3%, and 2 are comparable to BF16. Similarly, for VLM, 2 out of 9 cases outperform BF16, 3 out of 9 underperform, and 4 are comparable. Overall, **COAT achieves accuracy on par with BF16**.
>
>
> ### **Q2**: How each component influences the model performance
>
> **A2**: This is a constructive suggestion! We separately quantize the optimizer and activations to better understand how each component influences the model performance. We find that **quantizing activation incurs almost no degradation due to our well-designed quantization flow**, and the **main degradation comes from optimizer quantization**. We choose to quantize both to maximize memory saving.
>
> |           | Train PPL      | COPA        | ARC (Easy) | SciQ | HellaSwag | Average |
> |-----------|----------------|-------------|------------|------|-----------|---------|
> | BF16      | 2.995          | 60.0        | 45.6       | 67.3 | 33.7      | 51.6    |
> | FP8 Optimizer + Activation | 3.008          | 61.0        | 44.2       | 67.6 | 33.7      | 51.5    |
> | FP8 Optimizer     | 2.998 (+0.003) | 60.0 (+0.0) | 44.5 (-1.1)       | 66.0 (-1.3) | 34.1 (+0.4)      | 51.1 (-0.5)    |
> | FP8 Activation     | 2.999(+0.004)  | 63.0 (+3.0) | 44.0 (-1.6)       | 68.4 (+1.1) | 34.1 (+0.4)     | 52.3 (+0.7)   |
>
> ### **Q3**: How/whether the dynamic range is synchronized across GPUs in the distributed training setting.
>
> **A3**: To clarify, the dynamic range is **NOT synchronized** across GPUs. COAT incurs no communication overhead. During distributed training, only gradients are communicated between GPUs while activations and optimizers stay on each GPU and are never sent across. Therefore, we calculate the dynamic range based on the statistics of the optimizer states on each device individually, eliminating the need for cross-GPU synchronization. For FSDP related implementation, we add more details in Section 3, lines 198-200 and Appendix A.

---

> ### Author Response · Authors · 2024-11-25
>
> Dear reviewer 4rZz,
>
> We are looking forward to further communication and please let us know if our response have fully addressed your concerns.
>
> Sincerely,
>
> Authors

---

> ### Author Response · Authors · 2024-12-02
>
> Dear Reviewer 4rZz,
>
> We truly appreciate your time and effort in reviewing our submission and offering valuable suggestions. The author-reviewer discussion period is closing soon, so we would like to confirm whether our responses have effectively addressed your questions and concerns.
>
> Thank you for your time and insights.
>
> Best regards,
>
> Authors

---

> ### Author Response · Authors · 2024-12-03
>
> Dear Reviewer 4rZz,
>
> Thank you again for your detailed suggestions and comments. As the deadline of discussion period is approaching, could you kindly review our responses and inform us whether you have further questions?
>
> Thank you!
>
> Sincerely,
>
> The Authors

---

### Author Response · Authors · 2024-11-23
**Response to all reviewers**

We thank all reviewers for the constructive advice and valuable feedback! We are happy that all reviewers appreciate the practical value of our work to the LLM community. Here, we would like to highlight the algorithm-wise contributions of COAT:

- We **identify the range under-utilization problem** for optimizers, which prevents the optimizer states from being quantized to FP8. While previous work [1] shows that naively quantizing optimizers leads to performance degradation, COAT's **dynamic range expansion** addresses this problem. It significantly reduces training memory by 4x while maintaining on-par performance, as shown in Table 3 / Figure 5.

- In addition to empirical experiments, we also **theoretically prove** that our polynomial expansion function is optimal for quantization (in Appendix E). This neat theory results translate into excellent training results—COAT's performance closely matches the BF16 baselines across various LLM and VLM tasks.

- COAT provides the first adaptive quantization flow for LLM training. It effectively reduces the quantization error by employing different quantization strategies for different layers, making it an important step to transform most training dataflows into FP8 without losing accuracy.

In summary,  COAT delivers significant value to the LLM community by enabling FP8 quantization to be effectively used in real-world training scenarios. Its contributions directly address the scalability and cost-efficiency challenges researchers and practitioners face by allowing large language models (LLMs) to train effectively on setups with fewer GPUs while also having a high training throughput. We have also released our codebase to anonymous GitHub to benefit the community (https://anonymous.4open.science/r/COAT-Anonymous-E13B/).

[1] Peng, Houwen, et al. "Fp8-lm: Training fp8 large language models." arXiv preprint arXiv:2310.18313 (2023).

## Proof the expand function is optimal

We prove that the choice of $f(x) = x^k$ is optimal given certain assumptions for the expansion function $f(x)$.

1. **$f(x)$ is an odd function**, where $f(x) + f(-x) = 0$, since the FP8's range is symmetric. Therefore, in the following discussion, we only consider $x \geq 0$.
2. **$f(x)$ is continuously differentiable.**
3. **$f(x)$ is monotonically increasing.** This is because we want to keep the relative order before and after quantization or expansion.
4. **$f(0) = 0$ and $f(1) = 1$.** $f(0) = 0$ since $f(x)$ is odd. $f(1) = 1$ since we can always consider $g(x) = \frac{f(x)}{f(1)}$ without changing its expansion ability.
5. **For any $x > y > 0, r > 0$, $\frac{f(x)}{f(y)} = \frac{f(rx)}{f(ry)}$.** This is used to make sure the function is scale-invariant. For example, if you multiply the unquantized value $X$ by $10$, the value after expansion and quantization is the same as if you do not multiply the value by $10.$

We prove that based on these assumptions, the only function that meets these assumptions is $f(x) = \text{sign}(x) |x|^k$.

### Proof
We only consider the case when $x > 0$. We denote $f(2) = m > 1$. From assumption 5, we have for any $p > r > s > q$, if $pq = rs$, then $f(p)f(q) = f(r)f(s)$, since
$ \frac{f(p)}{f(r)} = \frac{\big(p \cdot (s / p)\big)}{f\big(r \cdot (s / p)\big)} = \frac{f(s)}{f(q)}. $
And for any $p = rs$,
$
f(p) = f(p)f(1) = f(r)f(s).
$
Therefore, for any $x = 2^t, t \in \mathbb{N}$,
$$
f(x) = f(2^t) = f(2)^t = m^t.  (1)
$$
It is also true that for any $x = 2^{-t}, t \in \mathbb{N}$,
$$f(x) = \frac{f(1)}{f(\frac{1}{x})} = \frac{f(1)}{f(2^t)} = \frac{1}{m^t} = m^{-t},\text{ so that } f(2^{-t}) = m^{-t}.  (2)$$
Therefore, for any $N \in \mathbb{N}$, $x = 2^t$, $t = \sum\limits_{i=-N}^{N} c_i \cdot 2^i$, $c_i \in \{0, 1\}$,
$$
f(x) = f\left(\prod_{i=-N}^{N} 2^{c_i \cdot 2^i}\right) = \prod_{i=-N}^{N} f(c_i \cdot 2^i) = \prod_{i=-N}^{N} m^{c_i \cdot 2^i} = m^{\sum\limits_{i=-N}^{N} c_i \cdot 2^i} = m^t.  (3)
$$
For $\forall x > 0$, it can be written in the form of $x = 2^t, t\in \mathbb{R}$. Consider the binary representation of $t$, denoted by $t = \sum\limits_{i=-\infty}^{\infty} c_i \cdot 2^i$, where $c_i \in \{0, 1\}$.
For any integer $N > 0$, there exist two series $a_i, b_i, -N \leq i \leq N$ that satisfy
$a_{-N} = 0, b_{-N} = 1, \text{ and } a_i = b_i \text{ for } -N+1 \leq i \leq N,$
$$
t_a \leq t < t_b,
$$
where $t_a = \sum\limits_{i=-N}^{N} a_i \cdot 2^i$ and $t_b = \sum\limits_{i=-N}^{N} b_i\cdot 2^i$.
Since $f(x)$ is continuous, the values of $f(x)$ at $2^{t_a}$ and $2^{t_b}$ can be used to approximate $f(x) = f(2^t)$:
$$
f(x) = \lim_{N \to \infty} f(2^{t_a}) = \lim_{N \to \infty} f(2^{t_b}).  (4)
$$
Therefore, for $\forall~ t \in \mathbb{R}$, $f(2^t) = m^t$ for some $m > 0$. Therefore, $\forall~ x > 0$, $f(x) = m^{\log_2(x)}$.
Using the exponential and logarithmic identity $m^{\log_2(x)} = x^{\log_2(m)}$, this simplifies to:
$$
f(x) = x^{\log_2(m)},  (5)
$$
where $f(2) = m$ is a constant. If we denote $\log_2(m) = k$, then $f(x) = x^k$.

---

> ### Comment · Reviewer_P9H9 · 2024-11-23
>
> Thank you for providing this proof. I have a few comments:
> - The argument for $x=2^{-t}$ mistakenly uses $x^t=1$, but $x^t=2^{-t^2}$. Though this can be easily fixed as follows: $f(x) = \frac{f(x)}{f(1)} = \frac{f(x x^{-1})}{f(x^{-1})} = f(x^{-1})^{-1}$, and since $f(2^t)=m^t$, we have $f(2^{-t})=f(2^{t})^{-1} =  m^{-t}$.
> - The variables $x_a$ and $x_b$ were introduced without explanation. I believe the definitions for $t_a$ and $t_b$ were intended for them instead.
> - The claim that $f(x)=\text{sign}(x)|x|^k$ is the *only* function (or *optimal* function) that meets the assumptions should be made more precise (at least in the revision). What does optimality mean here? The claim may hold on $x \in \mathbb{R}$, but does this imply that it also holds on the constrained domain $x = \sum_{i=-N}^N a_i 2^i$ for all $a_i \in \{0, 1\}$ and finite $N$? This is just a technicality and I believe that the claim for the unconstrained case suffices for the purposes of this paper, but it could be the case that there exists a *better* (or another) $f$ for the constrained case.
>
> Also, thank you for providing the anonymous and professionally-written code.

---

> ### Author Response · Authors · 2024-11-23
>
> Thank you for the comment. We noticed some minor mistakes in our original proof. We have fixed the issues and updated our manuscript. **The conclusion remains the same.**
>
> Q1 & Q2: Thank you for pointing this out. The correct proof is updated in Equation (2):
>
> $f(x) = \frac{f(1)}{f(\frac{1}{x})} = \frac{f(1)}{f(2^t)} = \frac{1}{m^t} = m^{-t},\text{so that } f(x) = m^{-t}.$
>
> We intend to use $t_a$ and $t_b$ for this part. We also correct these terms.
>
> Q3: We also updated the proof after Equation (3) consequently. The binary representation should be expressed as $x = 2^t, t = \sum_{i=-\infty}^\infty c_i \cdot 2^i, c_i \in \{0, 1\}$. We replace $a_i$ with $c_i$ to avoid the repeated use of letters, and the binary representation should be discussed in terms of t.
>
> The intention of “optimality” was to argue that $f(x) = \operatorname{sign} (x) |x|^k$ is the only function that satisfies all the 5 assumptions on the unconstrained domain $x \in \mathbb{R}$.
>
> On the constrained domain where $x = 2^t, t = \sum_{i=-N}^N c_i \cdot 2^i, c_i \in \{0, 1\}$, our claim also holds. Equation 3 has already demonstrated that in this constrained domain, we have $f(2^t) = m^t$, where $m = f(2)$. This means that the only function on this constrained domain that satisfies all the 5 assumptions should have $f(x) = x^k, where k = \log_{2}{m}$. This proves the optimality of this function.

---

### Meta-Review · Area_Chair_bsoV · 2024-12-15

**Metareview:**

This paper introduces COAT, an FP8 training framework that improves memory efficiency and training speed for large-scale models while maintaining BF16-level performance. Key contributions include dynamic range expansion, aligning optimizer states to FP8 value ranges, and mixed-granularity activation quantization, combining per-tensor and per-group strategies. While the techniques are incremental and lack substantial novelty, their practical significance is clear, with strong experimental results and potential for adoption by practitioners. The paper could benefit from more detailed ablations and comparisons with additional relevant related works as requested by reviewers. Overall, I recommend borderline acceptance for its practical contributions to efficient training.

**Additional Comments On Reviewer Discussion:**

All the reviewers recommend the paper for weak acceptance. None of the reviewers is particularly excited about the paper's contribution. However, they all agree that the contribution is interesting and worth publication.

---

### Decision · Program_Chairs · 2025-01-22

Accept (Poster)